# Sintering-induced cation displacement in protonic ceramics and way for its suppression

Ze Liu[1,2], Yufei Song[3], Xiaolu Xiong[1], Yuxuan Zhang[1], Jingzeng Cui[1,2], Jianqiu Zhu[1,2], Lili Li[4], Jing Zhou[1], Chuan Zhou[5], Zhiwei Hu[6], Guntae Kim[1], Francesco Ciucci[7], Zongping Shao[8] ✉, Jian-Qiang Wang[1,2] ✉ & Linjuan Zhang[1,2] ✉

Protonic ceramic fuel cells with high efficiency and low emissions exhibit high potential as next-generation sustainable energy systems. However, the practical proton conductivity of protonic ceramic electrolytes is still not satisfied due to poor membrane sintering. Here, we show that the dynamic displacement of $Y^{3+}$ adversely affects the high-temperature membrane sintering of the benchmark protonic electrolyte $BaZr_{0.1}Ce_{0.7}Y_{0.1}Yb_{0.1}O_{3-\delta}$, reducing its conductivity and stability. By introducing a molten salt approach, pre-doping of $Y^{3+}$ into A-site is realized at reduced synthesis temperature, thus suppressing its further displacement during high-temperature sintering, consequently enhancing the membrane densification and improving the conductivity and stability. The anode-supported single cell exhibits a power density of $663\ mW\ cm^{-2}$ at 600 °C and long-term stability for over 2000 h with negligible performance degradation. This study sheds light on protonic membrane sintering while offering an alternative strategy for protonic ceramic fuel cells development.

Protonic ceramic fuel cells (PCFCs), with a low activation energy and no oxidation-product dilution of the fuel gas, exhibit high potential as next-generation ceramic fuel cells with lower operating temperature and higher energy-conversion efficiency compared with conventional solid oxide fuel cells (SOFCs) based on oxygen ion-conducting yttrium stabilized zirconia (YSZ) electrolyte[1–4]. During the past several years, a quickly expanded interest and research activities in PCFCs have been envisioned[5–9].

Despite the great promises, the advance in cell performance of PCFCs is still far behind the conventional SOFCs based on oxygen ion-

conducting electrolytes[10]. One of the main reasons is the big challenge in the fabrication of dense protonic electrolytes with high conductivity and stability[11–13]. Most of the protonic ceramic electrolytes are barium-contained perovskite oxides, such as $BaZr_{0.1}Ce_{0.7}Y_{0.2}O_{3-\delta}$ and $BaZr_{0.1}Ce_{0.7}Y_{0.1}Yb_{0.1}O_{3-\delta}$ (BZCYYb)[14–16]. It is well known that barium evaporates easily at high temperature. During the sintering for electrolyte densification, which is usually higher than 1350 °C, the loss of barium from evaporation becomes not negligible, bringing a negative effect on the electrolyte sintering and conductivity[17–20]. Therefore, excessive barium is typically applied during the high-temperature

[1]Key Laboratory of Interfacial Physics and Technology, Shanghai Institute of Applied Physics, Chinese Academy of Sciences, Shanghai 201800, China. [2]University of Chinese Academy of Sciences, Beijing 100049, China. [3]Department of Mechanical and Aerospace Engineering, The Hong Kong University of Science and Technology, Clear Water Bay, Hong Kong, China. [4]State Key Laboratory of Crystal Materials and Institute of Crystal Materials, Shandong University, Jinan 250100, China. [5]State Key Laboratory of Materials-Oriented Chemical Engineering, College of Chemical Engineering, Nanjing Tech University, Nanjing 211816, China. [6]Max Planck Institute for Chemical Physics of Solids, Dresden 01187, Germany. [7]Chair of Electrode Design for Electrochemical Energy Storage Systems, University of Bayreuth, Weiherstraße 26, Bayreuth 95448, Germany. [8]WA School of Mines: Minerals, Energy and Chemical Engineering, Curtin University, Perth WA6845 WA, Australia. ✉e-mail: zongping.shao@curtin.edu.au; wangjianqiang@sinap.ac.cn; zhanglinjuan@sinap.ac.cn

sintering to compensate for the potential barium loss from evaporation[21]. The mechanism of how the barium evaporation deteriorate the membrane sintering is still not clear. Although barium excess does occasionally bring beneficial effects for electrolyte densification, the improvement is usually marginal[22,23]. It suggests, in addition to the barium loss, that some other factors could also play an important role in the protonic perovskite sintering and, consequently, the ionic conductivity.

Up to now, the perovskites with the highest protonic conductivity are always based on $BaCeO_3$ and $BaZrO_3$ parent oxides, and Grotthuss mechanism is usually accepted to describe the proton migration in such electrolyte materials[24–28]. Through the Grotthuss mechanism, proton is introduced into the oxide lattice through the hydration of oxygen vacancies, then the proton is hopping between neighboring oxygen ion sites. The oxygen vacancies are introduced through the doping of Ce and Zr sites in $BaCeO_3$ and $BaZrO_3$, respectively, with a lower oxidation state cation(s)[29,30]. To meet the tolerance factor requirement for a stable perovskite lattice, the ionic radius of such dopants should be similar to those of the hosts, so $Y^{3+}$ and $Yb^{3+}$ are the most frequently used as the dopants[31,32]. Interestingly, $Y^{3+}$ could also be doped into the A-site of perovskites. Actually, there are many reports about the A-site $Y^{3+}$ doped perovskites in literature[18,33,34].

In this study, we report the presence of cation displacement between A-site and B-site of protonic perovskites during high-temperature sintering processes, which significantly influences the electrolyte densification and its conductivity. Specifically, the $Y^{3+}$, which is designed on purpose to the perovskite B-site of BZCYYb, could be dynamically entered into the A-site ($Ba^{2+}$) during the high-temperature sintering, resulting in a lattice shrinkage, which deteriorates the electrolyte densification. The displacement of $Y^{3+}$ from the perovskite B-site to the A-site also causes a reduction in oxygen vacancy concentration of the material. Both these factors adversely affect the electrolyte conductivity. We further propose a molten salt synthesis (MSS) method using KCl and NaCl as molten salt media to fabricate BZCYYb electrolyte, which can effectively suppress such dynamic cation displacement during high-temperature sintering. The micron-scale monodisperse BZCYYb with uniform particle-size distribution and moderate packing density as synthesized shows a reduced number of enclosed pores within the green electrolyte and enhanced sintering. The as-obtained BZCYYb electrolyte membrane shows high proton conductivity, reaching $4.7 \times 10^{-2}$ S cm$^{-1}$ at 600 °C, which is 5.2 times that of the sample prepared by the conventional solid-state reaction (SSR) method. The corresponding anode-supported single cell with BZCYYb electrolyte synthesized by the MSS method exhibits low ohmic resistance and distinguished peak power density (PPD) of 964 mW cm$^{-2}$ at 650 °C. The cell demonstrates excellent electrochemical stability without obvious degradation in a total operation period of over 2000 h at 600 °C. Therefore, this study enables an in-depth understanding of protonic electrolyte sintering and proposes a new strategy for developing high-performance PCFCs with a wide range of potential applications.

## Results

### Dynamic $Y^{3+}$ displacement and its detrimental effect

First, to investigate dynamic cation displacement in protonic ceramics, a BZCYYb sample with stoichiometric cation compositions was prepared by the conventional solid-state reaction method by calcining the solid precursor at 1000 °C for 10 h in air, denoted as BZCYYb-SSR. According to the room temperature X-ray diffraction (XRD) patterns, the as-synthesized BZCYYb-SSR sample possesses a single-phase perovskite structure (space group: *Imma*, $a = 6.21$ Å, $b = 8.81$ Å, $c = 6.22$ Å) (Supplementary Fig. 1 and Supplementary Table 3)[35]. Moreover, the scanning electron microscopy (SEM) images and particle-size distribution of BZCYYb-SSR reveals irregular and significantly agglomerated particles with an average particle size of 1.95 μm and a specific

surface area of 1.90 m$^2$ g$^{-1}$ (Supplementary Figs. 2–4). It is generally believed that barium is easily evaporated at high temperature, which could deteriorate the sintering of the electrolyte membrane and consequently cause poor conductivity. The energy dispersive spectroscopy (EDS) and high-resolution inductively coupled plasma-mass spectrometry (HR-ICP-MS) were then conducted to determine the elemental composition of the as-synthesized BZCYYb-SSR. The results show that the Ba content in the sample is almost equal to the theoretical value as designed (1.01 vs. 1) (Supplementary Fig. 5 and Supplementary Tables 1 and 2), thus indicating negligible barium loss from evaporation during the calcination at 1000 °C.

For PCFC application, the electrolyte should be fabricated into dense membranes, thus requiring sintering at high temperatures, sometimes even surpassing 1400 °C[3]. To determine the phase structure of the material at different temperatures, we first annealed the as-synthesized BZCYYb-SSR at certain temperatures (1100, 1200, 1300, 1400 and 1450 °C) in air for 10 h, and quenched to room temperature to maintain the high-temperature crystal structure, which was then subjected for room temperature XRD analysis. As shown in Fig. 1a and Supplementary Fig. 6, when the samples were quenched from temperatures ≥1200 °C, their XRD peaks exhibited an obvious shift in position as compared to the pristine sample without the treatment. Typically, due to the thermal excitation, an expansion of the lattice should be expected with the increase of treatment temperature. Interestingly, the shift of diffraction peaks to higher angles was observed at high quenching temperatures, indicating severe lattice shrinkage (Supplementary Figs. 6–8 and Supplementary Table 3). In other words, the BZCYYb-SSR exhibits a negative thermal expansion coefficient feature at high temperature.

To determine if such severe lattice shrinkage of BZCYYb-SSR was caused by A-site deficiencies owing to Ba evaporation during annealing, A-site deficient $Ba_{0.98}Zr_{0.1}Ce_{0.7}Y_{0.1}Yb_{0.1}O_{3-\delta}$, $Ba_{0.95}Zr_{0.1}Ce_{0.7}Y_{0.1}Yb_{0.1}O_{3-\delta}$, $Ba_{0.93}Zr_{0.1}Ce_{0.7}Y_{0.1}Yb_{0.1}O_{3-\delta}$ were also synthesized using the SSR method by calcination of the corresponding precursors at 1000 °C in air, denoted as BZCYYb-0.98-SSR, BZCYYb-0.95-SSR, BZCYYb-0.93-SSR, respectively (Supplementary Figs. 9–20). For these samples, some impurity $Y_2O_3$ phase was detected, suggesting the difficulty of obtaining the phase-pure A-site deficient samples[18]. Indeed, the presence of A-site deficiencies led to the shrinkage of the lattice, for example, the cell volumes of BZCYYb-1.0, 0.98, 0.95 and 0.93 are 340.3, 339.6, 339.5, and 337.4 Å$^3$, respectively, at room temperature (Supplementary Table 3). However, when all four samples were annealed at 1300 °C, their diffraction peak positions became similar to each other (Supplementary Fig. 21), suggesting the barium evaporation may not be the sole reason for the shrinkage of BZCYYb-SSR lattice after the sintering at 1300 °C and higher temperatures since these four samples possessed different barium contents. From the XRD refinement results, the $a$ and $c$ axes of all the samples underwent significant shrinkage after quenching (Supplementary Table 3). In the perovskite structure, the $a$ and $c$ axes correspond to the A−O and B−O plane layers, respectively, and the contraction of the $a$ and $c$ axes indicates the shortening of the A−O and/or B−O bond lengths (Supplementary Fig. 22). According to HR-TEM, the BZCYYb-SSR and BZCYYb-0.93-SSR samples calcined at 1000 °C showed the interplanar distance of 0.317 and 0.312 nm, respectively, corresponding to the (002) crystal plane (Fig. 1c and Supplementary Figs. 23 and 24). The BZCYYb-SSR and BZCYYb-0.93-SSR samples annealed at 1450 °C show significantly decreased interplanar distances relative to the as-synthesized samples (Fig. 1c and Supplementary Figs. 25–28), confirming the severe lattice shrinkage, consistent with the XRD results. We also demonstrated that even after increasing the stoichiometry of Ba to 1.05 (BZCYYb-1.05-SSR), the sample underwent significant lattice shrinkage during high-temperature sintering (Supplementary Figs. 29-32 and Supplementary Table 4), which implies that Y migration cannot

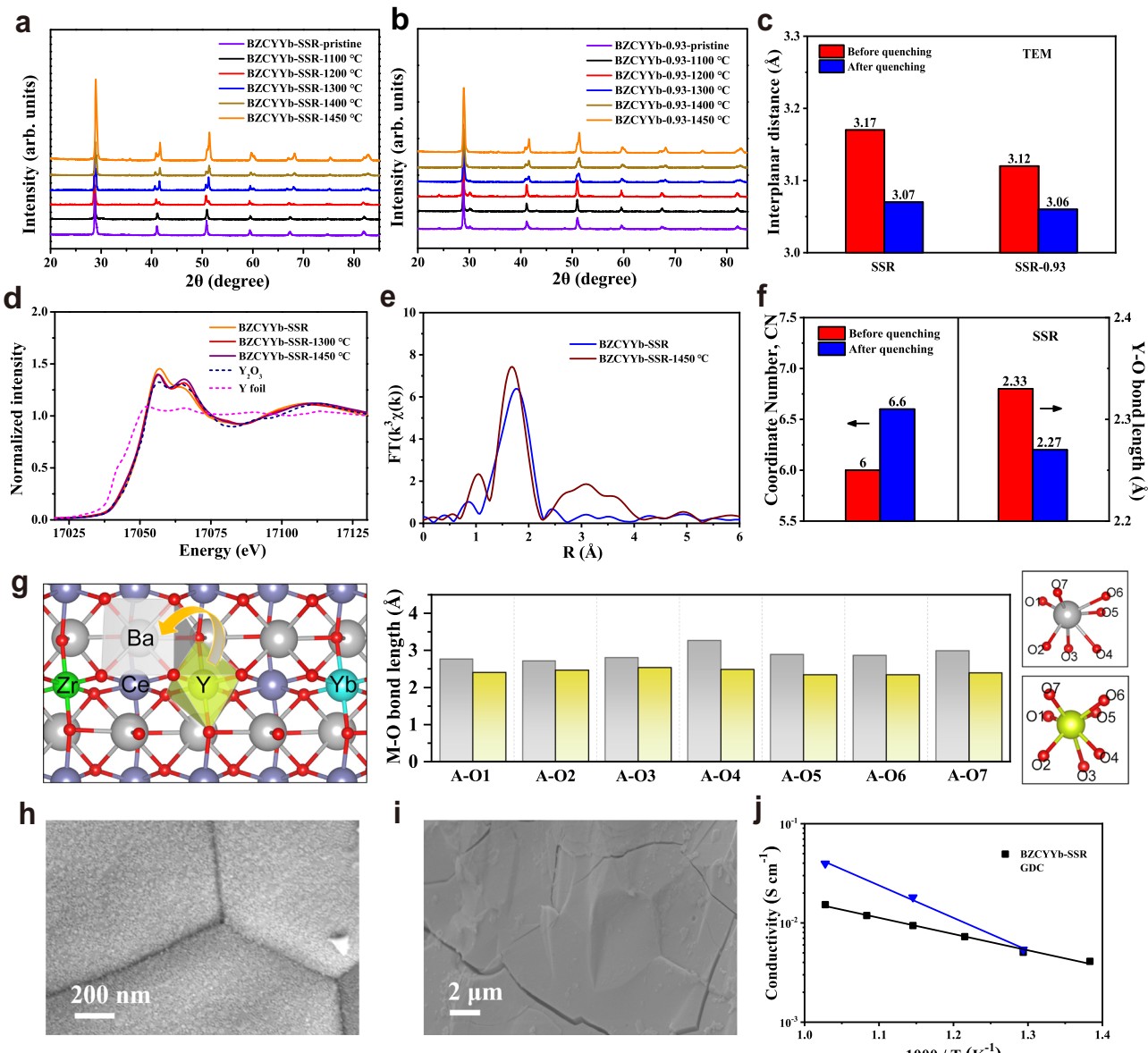

**Fig. 1 | Study of structural evolution of BZCYYb prepared by conventional SSR method. a**, **b** XRD patterns of pristine BZCYYb-SSR and BZCYYb-0.93-SSR samples and after quenching at different temperatures. **c** The change of lattice interplanar spacing of (002) crystal plane before and after quenching at 1450 °C. **d** The spectra of Y $K$-edge XANES for BZCYYb-SSR before and after quenching at 1450 °C. **e** Fourier-transformed of Y $K$-edge EXAFS spectra of BZCYYb-SSR before and after quenching at 1450 °C. **f** The coordinate number and Y−O bond length for BZCYYb-SSR before and after quenching at 1450 °C. **g** Diagram of the Y occupying the A-site (Ba) in perovskite, and the corresponding bond length with the neighboring oxygen atoms before and after Y substitution. Surface (**h**) and cross-sectional (**i**) SEM images of BZCYYb-SSR membrane sintered at 1450 °C. **j** Temperature dependence of conductivity for BZCYYb-SSR and GDC[37] membrane.

be inhibited by increasing the stoichiometry of Ba for BZCYYb-SSR samples.

To understand the origin of such obvious lattice shrinkage, the local atomic coordination environment and electronic structure of BZCYYb before and after annealing at 1450 °C were characterized by X-ray absorption spectroscopy (XAS). Figure 1d shows the X-ray absorption near edge structure (XANES) at the Y $K$-edge for pristine and annealed BZCYYb-SSR. The valence states of Y cation in all BZCYYb samples are close to +3. The Fourier-transformed extended X-ray absorption fine structure (FT-EXAFS) spectra of BZCYYb have been analyzed at the Y $K$-edge to track changes in the coordination shell of $Y^{3+}$ (Fig. 1e, Supplementary Figs. 33–36, and Supplementary Table 5). According to EXAFS fitting, the average Y−O distance of the BZCYYb-SSR before and after the annealing at 1450 °C is $2.33 \pm 0.01\,\text{Å}$ vs. $2.27 \pm 0.01\,\text{Å}$, respectively, thus confirming the severe lattice shrinkage

after the annealing. In perovskite structures, the A-site cations usually coordinate with 12 oxygen atoms, while the B-site cations coordinate with 6 oxygen atoms. The coordination configurations of Y of these samples were further examined by EXAFS fitting (Supplementary Table 5). The coordination number of Y in the BZCYYb-SSR calcined at 1000 °C in air for 10 h is 6. It suggests that all the $Y^{3+}$ was located at the B-site of BZCYYb-SSR as designed. However, after the annealing at 1450 °C, the coordination number of Y in BZCYYb-SSR sample increased dramatically to 6.6, suggesting partial $Y^{3+}$ was displaced from B-site to A-site, which is likely facilitated by the A-site cation deficiency as caused by the Ba evaporation. Since $Y^{3+}$ in A-site is smaller than $Ba^{2+}$ and $Y^{3+}$ at B-site is larger than $Zr^{4+}$, $Ce^{4+}$ and $Yb^{3+}$, and the displacement of $Y^{3+}$ from B-site to A-site caused the shrinkage of the perovskite lattice[32,36]. It was observed that $Yb^{3+}$ was preferentially precipitated in the form of $Yb_2O_3$ phase during the high-temperature

sintering process (Supplementary Figs. 37 and 38). Moreover, X-ray photoelectron spectroscopy (XPS) results indicate that the lattice oxygen content increased obviously after the quenching, which means that the oxygen vacancy content in the system decreased, this is in well agreement with the fact that the lattice rearrangement because of the displacement of $Y^{3+}$ from B-site to A-site would consume the oxygen vacancy (Supplementary Fig. 39).

Subsequently, we performed density functional theory (DFT) simulations to confirm the displacement of $Y^{3+}$ from B-site to A-site of BZCYYb-SSR during high-temperature sintering (Supplementary Fig. 40 and Supplementary Table 6). There are two potential ways for the displacement. One case is that Y enters into the Ba vacancy with the Ba evaporation or in the form of Ba oxides in the grain boundary after quenching. For this case, both Y–O bond length and cell volume should be reduced, this is consistent with the experimental results (Fig. 1g, Supplementary Fig. 41, and Supplementary Tables 7 and 8). The other case is that Y enters into A-site and Ba enters into B-site, forming an anti-phase defect structure. Based on the calculations, both the average Y–O bond length and cell volume should be increased, which is contrary to the experimental results (Supplementary Fig. 41 and Supplementary Table 9). Therefore, the DFT simulations further confirm that the lattice shrinkage of BZCYYb-SSR after the quenching can be attributed the displacement of $Y^{3+}$ from B-site to A-site. To clarify $Y^{3+}$ displacement, DFT calculations have also been performed involving the corresponding configurations before and after Y diffusion (Supplementary Fig. 42 and Supplementary Table 10). More stable configurations can be found after Y diffusion from B-site to A-site, indicating that the diffusion of Y is favored from thermodynamical view.

The aforementioned results indicate that BZCYYb samples prepared by the conventional SSR method undergo severe dynamic $Y^{3+}$ displacement at high sintering temperatures (>1300 °C), leading to lattice shrinkage. Typically, high sintering temperatures (>1350 °C) are used to fabricate dense electrolyte membranes for PCFCs. It means that dynamic $Y^{3+}$ displacement is inevitable during the PCFCs fabrication for the sample prepared by the SSR method, while such displacement would cause the rearrangement of the lattice, which could have a crucial effect on the membrane densification and, consequently, the conductivity. In this study, electrolyte membranes sintered at 1450 °C in air for 5 h were used to investigate the potential effect of the dynamic $Y^{3+}$ displacement in BZCYYb on the electrolyte densification and proton conductivity. To evaluate the density of BZCYYb-SSR, SEM was used to investigate the morphology of BZCYYb-SSR pellets sintered at 1450 °C (Fig. 1h, i and Supplementary Figs. 43 and 44). From the surface and cross-sectional views of the BZCYYb-SSR pellet, the electrolyte demonstrates rich pores and cracks in bulk, showing a relative density of 97.8%. In combination with the XRD results, it is likely that, during the sintering of BZCYYb-SSR pellet, the Ba evaporation and Y displacement caused large lattice distortion and strain, thus leading to a low densification and the formation of cracks. Consequently, the BZCYYb-SSR membrane shows an inferior conductivity from $0.5 \times 10^{-3}$ to $1.5 \times 10^{-2}$ S cm$^{-1}$ at the temperature range between 450 and 700 °C, which is even lower than that of the reported oxygen ion conduction electrolyte, such as $Ce_{0.8}Gd_{0.2}O_{1.9}$ (GDC)[37] (Fig. 1j).

## Suppressing the dynamic $Y^{3+}$ displacement and benefits

To suppress the dynamic $Y^{3+}$ displacement during PCFC fabrication, it is important to pre-dope the Y into the A-site of perovskite. Unfortunately, based on the conventional solid-state reaction method, it was found such Y doping into the A-site only happens at calcination of higher than 1200 °C, while the over-calcination of the electrolyte powder would seriously coarsen the particles, thus reducing the sintering capability of the membrane. The MSS method proposed in this

study enabled the pre-doping of Y into the A-site of BZCYYb at relatively low temperatures. As seen in Supplementary Fig. 45, at the salt-to-nitrate ratios of 2/1, single-phase BZCYYb-MSS was successfully formed after the calcination at 800–900 °C. For comparison, BZCYYb-MSS samples were synthesized using salt-to-nitrate ratios of 1/1, 3/1, and 4/1 (Supplementary Figs. 46 and 47). Figure 2a indicates that the BZCYYb-MSS synthesized using a salt-to-nitrate ratio of 2/1 shows the most uniform particle-size distribution among all the samples, with an average particle size of 1.86 µm and specific surface area of 3.36 m$^2$ g$^{-1}$ (Supplementary Figs. 48–50). Hence, BZCYYb-MSS synthesized with a salt-to-nitrate ratio of 2/1 was used for subsequent experimentation. According to XRD refinement, the BZCYYb-MSS shows an orthorhombic perovskite structure (space group: *Imma*, $a = 6.21$, $b = 8.74$, $c = 6.22$) with the reasonable reliability fitting factor of 1.84 (Supplementary Fig. 51 and Supplementary Table 3).

To further investigate the composition of the BZCYYb synthesized by the MSS method, EDS and HR-ICP-MS were conducted. According to the EDS mapping and HR-ICP-MS results, the Ba content of BZCYYb-MSS is smaller than the designed stoichiometry (0.93 vs. 1) (Supplementary Fig. 52 and Supplementary Tables 1 and 2). Based on the SSR experiments, as mentioned before, the loss of barium due to evaporation is unlikely at temperatures lower than 1000 °C; therefore, additional barium is expected to be present in the molten salt during synthesis, which is washed out during the washing process. The TEM images shown in Fig. 2b–d indicate a well-defined crystalline structure with a distinct grain edge, the selected area electron diffraction (SAED) pattern confirms the long-range order in the crystal structure, whereas the HR-TEM image indicates a highly crystalline nature corresponding to the (002) crystal plane of BZCYYb with interplanar distance of 0.315 nm.

For the BZCYYb-MSS samples, quenching treatments at the same temperature as BZCYYb-SSR were also carried out. As seen in Fig. 2e, the diffraction peaks of BZCYYb-MSS hardly changed despite the temperature rising to 1450 °C. Thus, the as-synthesized BZCYYb-MSS sample maintained a stable crystal structure during high-temperature sintering (Supplementary Figs. 53–55 and Supplementary Table 3). The lattice spacing of (002) crystal plane as obtained from HR-TEM (Fig. 2f) also suggests a negligible lattice shrinkage in the BZCYYb-MSS during the sintering (Supplementary Figs. 56 and 57), which further demonstrates the superior structural stability. The EXAFS fitting of the BZCYYb-MSS samples before and after quenching at 1450 °C give the Y–O distances of $2.31 \pm 0.01$ and $2.29 \pm 0.01$ Å, respectively, confirming the negligible lattice shrinkage (Fig. 2g, h and Supplementary Figs. 58–60). This differs notably from the corresponding BZCYYb-SSR samples. The EXAFS fitting results demonstrate that the coordination numbers of the Y in BZCYYb-MSS samples before and after quenching at 1450 °C are 6.3 and 6.4, respectively. This result indicates that in the as-synthesized BZCYYb-MSS sample at a relatively low temperature (800–900 °C), partial Y cation was already doped into the A-site of the perovskite lattice, which effectively suppressed the further displacement of B-site $Y^{3+}$ to the A-site during high-temperature (1300 °C or higher) sintering. And we proved that the fine structure of Ce did not change before and after quenching (Supplementary Fig. 61). In addition, XPS results also indicate that the oxygen vacancy concentration did not obviously change before and after quenching, thus confirming no obvious dynamic $Y^{3+}$ displacement in BZCYYb-MSS during high-temperature sintering (Supplementary Fig. 62). As to the SSR, even we introduced Ba deficiency in BZCYYb, $Y^{3+}$ displacement was not happened during powder synthesis, instead, a $Y_2O_3$ secondary phase was formed in BZCYYb-0.93-SSR to reduce the A-site cation deficiency (Supplementary Fig. 63). We further measured the cation compositions of BZCYYb-MSS and BZCYYb-SSR after quenching at different temperatures by HR-ICP-MS and calculated the percentage of Y occupying A- and B-sites in perovskite based on the FT-EXAFS fitting

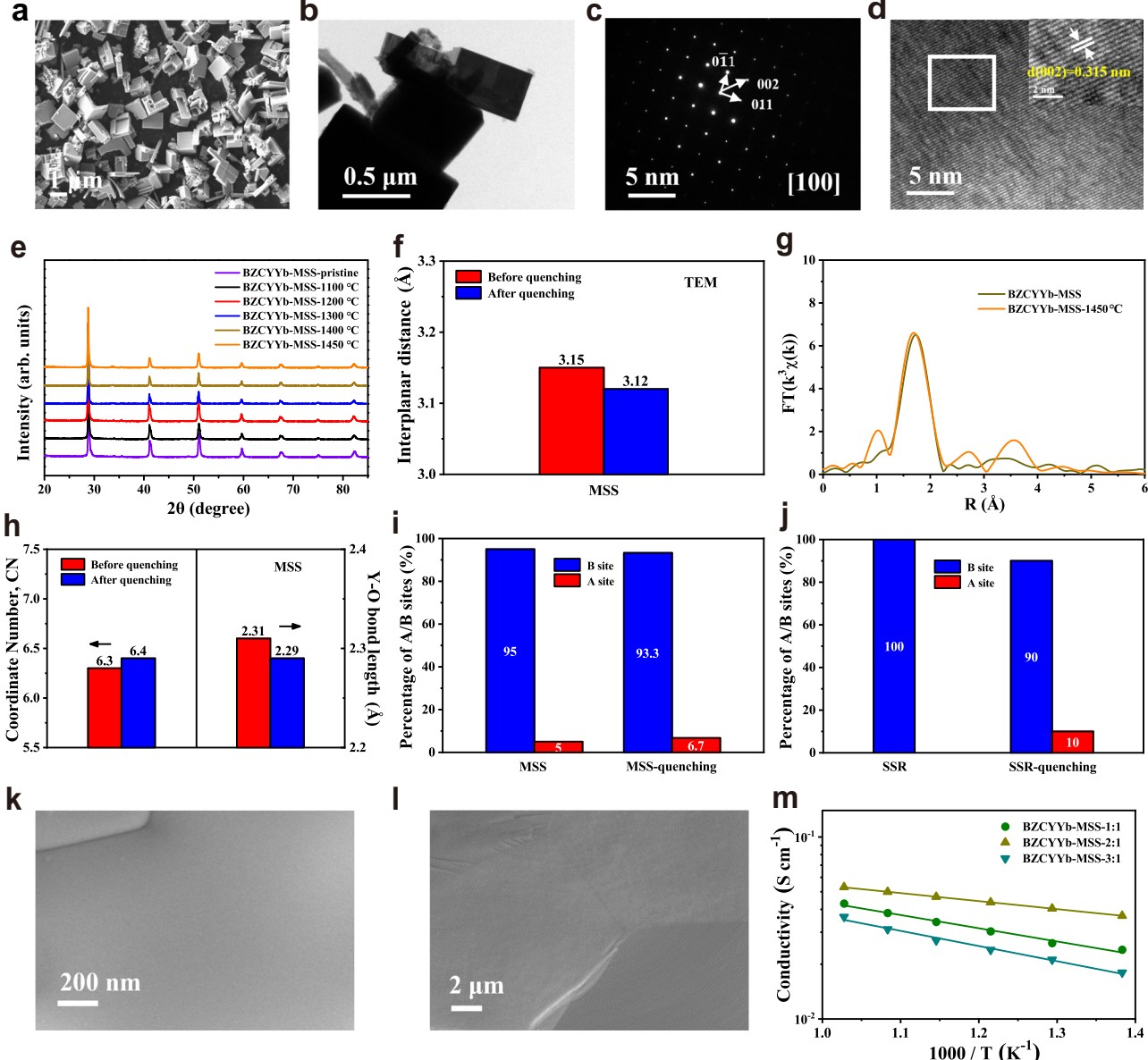

**Fig. 2 | Structural characterizations of BZCYYb prepared by MSS method. a** SEM image of BZCYYb-MSS. **b** TEM image of BZCYYb-MSS. **c** The SAED pattern and **d** HR-TEM images of BZCYYb-MSS from the marked area in (**b**), and the insert image is a higher magnification of the marked area in (**d**). **e** XRD patterns of pristine BZCYYb-MSS and after quenching at different temperatures. **f** The lattice interplanar spacing of (002) crystal plane before and after quenching at 1450 °C. **g** Fourier-transformed of Y $K$-edge EXAFS spectra of BZCYYb-MSS before and after quenching at 1450 °C. **h** The coordinate number and Y–O bond length for BZCYYb-SSR before and after quenching at 1450 °C. **i**, **j** Proportion of Y in sites A and B of BZCYYb-MSS and BZCYYb-SSR samples before and after quenching at 1450 °C. Surface (**k**) and cross-sectional (**l**) SEM images of BZCYYb-MSS membrane sintered at 1450 °C. **m** Temperature dependence of conductivity for BZCYYb-MSS membrane.

data (Fig. 2i, j and Supplementary Table 11). For the BZCYYb-MSS, the calculated perovskite formula of the as-synthesized sample is $(Ba_{0.93}Y_{0.005})(Zr_{0.1}Ce_{0.71}Y_{0.095}Yb_{0.1})O_{3-\delta}$, where 5% of the Y occupies the A-site. After quenching at 1450 °C, a small increase in the proportion of Y at A-site, about 6.7%, and the formula is $(Ba_{0.92}Y_{0.0067})(Zr_{0.08}Ce_{0.72}Y_{0.0933}Yb_{0.1})O_{3-\delta}$. As for the BZCYYb-SSR sample, after quenching at 1450 °C, the proportion of Y accommodates on A-site increases significantly from 0 to 10%, and the perovskite formula becomes $(Ba_{0.95}Y_{0.01})(Zr_{0.09}Ce_{0.71}Y_{0.09}Yb_{0.1})O_{3-\delta}$. We also demonstrated that BZCYYb prepared by the common sol-gel method, denoted as BZCYYb-SG, could not inhibit Y migration (Supplementary Figs. 64–67 and Supplementary Table 12). The suppressed dynamic $Y^{3+}$ displacement by the molten salt synthesis method would be beneficial for the membrane sintering since the structure rearrangement will be minimized during the sintering process.

To understand the impact of cation displacement on membrane sintering, BZCYYb-MSS pellets sintered at 1450 °C were analyzed by SEM. As expected, compared to BZCYYb-SSR membranes, the BZCYYb-MSS membranes show enhanced densification (98.4%), smooth surface and large grain size, with an average grain size of 2.31 μm, larger than that of BZCYYb-SSR (1.41 μm), and no cracks (Fig. 2k, l and Supplementary Figs. 68, 69, and 44). In addition, all the BZCYYb-MSS samples show higher protonic conductivities than BZCYYb-SSR samples (Fig. 2m). The BZCYYb-MSS sample synthesized using a molten salt ratio of 2:1 (BZCYYb-MSS-2:1) exhibits the highest proton conductivity with a value of $4.7 \times 10^{-2}$ S cm$^{-1}$ at 600 °C, as a comparison, it is $9 \times 10^{-3}$ S cm$^{-1}$ at the same temperature for BZCYYb-SSR. Moreover, the BZCYYb-MSS-2:1 sample also demonstrates the lowest electrical conduction activation energy (Supplementary Fig. 70).

The reasons for the high proton conductivity of BZCYYb-MSS membranes can be summarized as follows. First, for BZCYYb-SSR, dynamic $Y^{3+}$ displacement occurs due to Ba evaporation during the fabrication of dense membranes at high sintering temperatures (1450 °C) (Fig. 3a), which causes a large lattice distortion and strain, thus resulting in low densification and significant cracking of electrolyte, harmful to proton transport. However, for BZCYYb-MSS, the pre-doping of $Y^{3+}$ into the A-site during synthesis significantly suppresses dynamic $Y^{3+}$ displacement during high-temperature sintering (Fig. 3b), thus reducing lattice distortion and strain, leading to increased densification and reduced cracks, benefiting to proton transport. Second, the monodisperse and symmetrical microparticles of BZCYYb-MSS show a large surface area, which benefits solid-state diffusion, thereby further increasing the densification of electrolyte pellet and reducing the grain boundary for promoting proton transport. We also performed DFT calculations to investigate the proton transport between Y-substituted B-site and A-site (Supplementary Figs. 71 and 72). The energy barriers ($E_b$) for the Y-substituted B-site are 0.34 and 0.43 eV (Supplementary Fig. 73), whereas the energy barrier for the Y-substituted A-site is 0.33 eV (Supplementary Fig. 74). Clearly, these energy barriers are all quite small. Therefore, it is expected that proton migration can be achieved for different substitution sites of Y.

## The applicability of BZCYYb-MSS in PCFCs

Both the crystal structure and thermal expansion of electrolyte influence its proton conductivity and thermo-compatibility with other cell components, which finally determines the durability of PCFCs[38]. The crystal-structure stabilities of BZCYYb-MSS and BZCYYb-SSR after quenching at 1300 °C were investigated after heating both samples at 700 °C in air for 120 h (Supplementary Fig. 45). XRD patterns clearly confirm that the quenched powder derived from BZCYYb-SSR comprised a carbonate phase, whereas the quenched sample derived from BZCYYb-MSS maintained the perovskite structure without the formation of a secondary phase. HT−XRD was also employed to evaluate the crystal stability of BZCYYb-MSS under 5 vol% $CO_2$−Ar. No peaks corresponding to the new phase were detected on increasing the temperature from 150 to 650 °C, thus confirming the excellent crystal-structure stability of BZCYYb-MSS under $CO_2$ at the operating temperatures of PCFCs (Supplementary Fig. 76). Notably, the high crystal stability of BZCYYb-MSS can be attributed to the presence of $Y^{3+}$ at the A-site, which increases the entropy of the material, thus stabilizing the crystal structure. Furthermore, BZCYYb-MSS also shows a similar thermal expansion coefficient of $9.5 \times 10^{-6} K^{-1}$ to reported values[39] (Supplementary Fig. 77).

To further evaluate the predominance of BZCYYb-MSS, PCFCs comprising Ni-BZCYYb-MSS anodes, $PrBa_{0.8}Ca_{0.2}(Co_{0.95}Fe_{0.05})_2O_{6-\delta}$ (PBCCF5, ~10 μm) cathodes, and BZCYYb-MSS (~20 μm) as the electrolyte were fabricated to investigate the performance of BZCYYb-MSS (Fig. 4a). Notably, no phase reactions were detected between BZCYYb-MSS and PBCCF5 (Supplementary Fig. 78). Moreover, the open-circuit voltages (OCVs) of the BZCYYb-MSS-based cell are 1.05, 1.09, 1.10, and 1.11 V at 650, 600, 550, and 500 °C, respectively, which are higher than those of the BZCYYb-SSR-based cell (1.01, 1.05, 1.07, and 1.09 V at 650, 600, 550, and 500 °C, respectively) (Fig. 4b, c). The enhanced OCV could be attributed to the improved electrolyte density and reduced cracks of the BZCYYb-MSS membrane. In addition, the BZCYYb-MSS-based cell shows PPDs of 946, 663, 412, 260, 182, 116, and 71 mW cm$^{-2}$ at 650, 600, 550, 500, 450, 400, and 350 °C, respectively (Fig. 4b, d). As expected, the BZCYYb-SSR-based cell shows lower PPDs of 505, 410, 276, and 168 mW cm$^{-2}$ at 650, 600, 550, and 500 °C, respectively (Fig. 4c, d and Supplementary Table 13). According to the ohmic resistances of the two cells (Fig. 4e and Supplementary Fig. 79), the BZCYYb-MSS-based cell exhibits significantly reduced ohmic resistances as compared to the BZCYYb-SSR-based cell, which is the main

reason for the higher power output of the former. The lower ohmic resistances of the BZCYYb-MSS-based cell compared with that of the BZCYYb-SSR-based cell can be attributed to the enhanced proton conductivity, consistent with the proton conductivity results (Figs. 1j and 2m). It should also be noted that with the decrease of temperature, the ohmic resistances of the BZCYYb-MSS-based cell increase to a lesser extent than that of the BZCYYb-SSR-based cell, possibly owing to the lower electrical-conductivity activation energy of the former system, as mentioned previously.

Long-term durability test of the BZCYYb-MSS-based cell at 600 °C as shown in Fig. 4f, with a low degradation rate of 3% per 1000 h (or 30 mV/1000 h) at a constant current density of 222 mA cm$^{-2}$ for 1020 h. Contrarily, the voltage of BZCYYb-SSR-based cell decreased from 0.88 to 0.35 V within 90 h of stability testing. Furthermore, we maintained durability test on the BZCYYb-MSS-based cell at a constant current density of 444 mA cm$^{-2}$. Encouragingly, after long-term stability of 1020 h, the cell still maintained excellent stable performance at higher current density with negligible degradation, further confirming the superior durability of the BZCYYb-MSS-based cell. Electrochemical impedance spectroscopy (EIS) was measured for the BZCYYb-MSS-based cell during long-term stability testing (Fig. 4g). The ohmic resistance of the BZCYYb-MSS-based cell increased slightly during the first 200 h of cell operation and subsequently stabilized, whereas that of the BZCYYb-SSR-based-cell increased significantly from 0.29 to 0.73 Ω cm$^2$ during stability testing (Supplementary Fig. 80). The microstructures of the BZCYYb-MSS-based cell were characterized by Focused Ion Beam Scanning Electron Microscope (FIB-SEM) and EDS mapping. The porous electrode and BZCYYb-MSS electrolyte exhibit good adhesion without delamination and cracking after 2000 h of operation (Supplementary Figs. 81–84). Although SEM images of the BZCYYb-SSR-based cell after long-term operation confirmed good adhesion between the porous electrode and BZCYYb-SSR electrolyte, cracks were formed inside the electrolyte membrane (Supplementary Fig. 85). Hence, we can conclude that the BZCYYb-MSS-based cell exhibits excellent stability owing to the high structural stability of BZCYYb-MSS combined with a dense and crack-free electrolyte film.

## Discussion

In conclusion, we first specified that the BZCYYb electrolyte synthesized via the conventional SSR method undergo significant Ba evaporation during PCFCs fabrication owing to the high-temperature sintering (typically >1400 °C), which consequently leads to dynamic $Y^{3+}$ displacement from the perovskite B-site to the A-site. The severe Ba evaporation and dynamic $Y^{3+}$ displacement cause large lattice distortion and strain, thereby resulting in low densification and cracking in the electrolyte, harmful to proton transport of electrolyte and PCFCs durability. Consequently, to overcome this limitation, an innovative MSS method was used to synthesize BZCYYb electrolyte. Experiment results suggest that the Ba-site deficiency and partial Y doping into A-site during the early BZCYYb synthesis process, which significantly suppress the further Ba evaporation and Y displacement during high-temperature sintering. Consequently, the lattice distortion and strain of BZCYYb are significantly reduced, thereby improving the densification, and reducing the cracks of electrolyte for greatly enhanced proton transport and durability of PCFCs. Furthermore, the monodisperse and symmetrical microparticles of BZCYYb-MSS exhibit a large surface area, which promotes solid-state diffusion, thus increasing the densification of electrolyte pellet, reducing the grain boundary, and enhancing proton conduction. As a result, BZCYYb-MSS-based PCFCs achieved enhanced performance and durability due to the high proton conductivity and crack-free electrolyte. Therefore, this study sheds light on a fundamental understanding of protonic membrane sintering process while offering a promising alternative strategy for PCFC development.

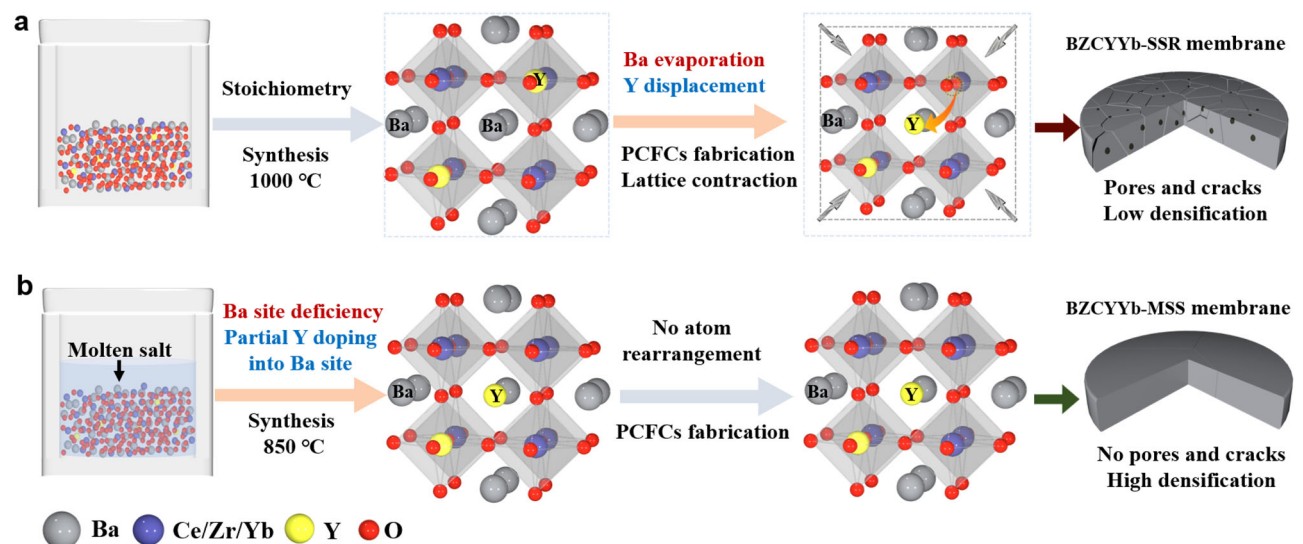

**Fig. 3 | Schematic diagrams of structural evolution. a** Schematic diagram of preparation and structural evolution of BZCYYb-SSR. **b** Schematic diagram of preparation of BZCYYb-MSS and inhibition of Y³⁺ displacement.

**Fig. 4 | Electrochemical performances of PCFC with BZCYYb electrolyte.**
**a** Cross-sectional SEM image of BZCYYb-MSS-based single cell after 2000 h long-term stability test. **b**, **c** I–V–P curves of BZCYYb-MSS and BZCYYb-SSR based single cell. **d** The comparison of the peak power densities of PCFCs with the two electrolytes. **e** The comparison of the ohmic resistance of the two electrolytes. **f** Long-term stability test of the two single cells at different constant current densities of 0.222 and 0.444 A cm⁻² at 600 °C. **g** Time dependence of ohmic resistances of the BZCYYb-MSS-based single cell at 600 °C.

## Methods

### Materials synthesis

$Ba_xZr_{0.1}Ce_{0.7}Y_{0.1}Yb_{0.1}O_{3-\delta}$ (BZCYYb) powders were fabricated by conventional solid-state reaction method. Stoichiometric amounts of $BaCO_3$, $ZrO_2$, $CeO_2$, $Y_2O_3$ and $Yb_2O_3$ were mixed by high-energy ball-milling in ethanol at 350 rpm for 10 h. The uniformly dispersed mixtures were then dried at 80 °C for 12 h and calcined at 1000 °C for 10 h in air. BZCYYb-MSS was synthesized by molten salt synthesis method, wherein KCl and NaCl were employed as molten salt at a molar ratio of 1:1. Stoichiometrically calculated amounts of $Ba(NO_3)_2$, $Ce(NO_3)_3 \cdot 6H_2O$, $Zr(NO_3)_4 \cdot 5H_2O$, $Y(NO_3)_3 \cdot 6H_2O$, $Yb(NO_3)_3 \cdot 5H_2O$ were mixed with different amounts of chloride salt, and then ground for several hours. The mixture was further calcined at 800–900 °C for 10 h. After calcining, the product was washed with deionized water, followed by filtration to remove the eutectic salt. The BZCYYb-MSS powder was obtained after repeated washing process and drying at 80 °C for 12 h. The air electrode powder of $PrBa_{0.8}Ca_{0.2}(Co_{0.95}Fe_{0.05})_2O_{6-\delta}$ (PBCCFe5) was fabricated by the commonly used sol-gel method. stoichiometrically $Pr(NO_3)_3 \cdot 6H_2O$, $Ba(NO_3)_2$, $Ca(NO_3)_2 \cdot 6H_2O$, $Co(NO_3)_2 \cdot 6H_2O$, and $Fe(NO_3)_3 \cdot 9H_2O$ were dissolved in deionized water. Using ethylenediaminetetraacetic acid (EDTA) and citric acid (CA) as complexing agents, with a molar ratio of 1:1:2 for total metal ions, EDTA and CA. A proper amount of $NH_3 \cdot H_2O$ was used to adjust the pH of aqueous solution to around 7. After by heating and stirring at 80 °C for 3 h to convert solution into viscous gel, the gel was moved to an oven and heated at 220 °C. The ash-like precursor was subsequently calcined in air at 1050 °C for 10 h to acquire pure perovskite powder.

### Fabrication of anode-supported cells

Anode-supported half-cells were prepared by dry-pressing and co-sintering process. The 0.35 g of anode raw precursor mixture consisting of NiO, BZCYYb, and soluble starch in a weight ratio of 6.5:3.5:1 was pressed at 75 MPa. 0.015 g of BZCYYb electrolyte powder was homogeneously distributed onto the surface of the green anode pellet, co-pressed at 150 MPa to form a thin electrolyte layer (about 20 μm after sintering), and then co-sintered at 1450 °C for 10 h. The PBCCF5 cathode with an effective area of 0.45 $cm^2$ was prepared by slurry spray deposition/sintering method. The slurry composed of PBCCF5 and a certain proportion of ethylene glycol, isopropanol and glycerol was sprayed onto the electrolyte surface, and finally sintered at 950 °C for 4 h.

### Physicochemical characterization

The crystal structures of BZCYYb powders were analyzed by X-ray diffraction (XRD, Bruker D8 Advance) using Cu $K_\alpha$ radiation in the range of 20–85°. The crystal lattice of the powders was investigated by transmission electron microscopy (TEM, Tecnai, G2 F20 S-TWIN). Microstructure and EDS mapping were investigated using focused ion beam scanning electron microscopy (FIB-SEM, Zeiss, Cross Beam 540). The cation compositions of samples were investigated by high-resolution inductively coupled plasma-mass spectrometry (HR-ICP-MS, Nu ATTOM). Thermal expansion coefficient of the sample was determined by dilatometer (Netzsch DIL 402 C) from 25 to 850 °C with a 10 °C $min^{-1}$ ramp rate under air atmosphere. Surface chemistry was analyzed by X-ray photoelectron spectroscopy using Thermo Scientific K-Alpha with Al $K_\alpha$ radiation.

### X-ray absorption spectroscopy measurements.

The X-ray absorption spectroscopy measurements at the Y $K$-edge were carried out in fluorescence mode at the BL14W1 beamline in Shanghai Synchrotron Radiation Facility (SSRF)[40]. All XAS data was analyzed following the standard procedures in the program Demeter[41]. $k^3$-Weighted Extended X-ray absorption fine structure oscillations were extracted from the normalized XAS spectra by subtracting the atomic background using a cubic spline fit, where $k$ is the photoelectron wave vector. The $k^3\chi(k)$ functions were then Fourier transformed into $R$-space, with the Hanning-type window in the range of 2.4–11.2 Å. Least-squares curve parameter fitting was performed to obtain the quantitative structural parameters around the Y atoms.

### Electrochemical measurements

Full cells were heated up to 650 °C with a heating rate of 3 °C $min^{-1}$ before testing. After reaching the operating temperature, wait for nickel oxide to be reduced to nickel metal under hydrogen atmosphere. EIS measurements were conducted at OCV conditions from 450–650 °C, with a frequency range from $10^6$ to $10^{-1}$ Hz, an a.c. amplitude of 10 mV. The electrochemical performances of current density–voltage curves from 450 to 650 °C and long-term durability test at 600 °C were collected by electrochemical working station (PGSTAT302N). For the single-cell test, the fuel electrode was exposed to 80 mL $min^{-1}$ dry hydrogen and the air electrode was exposed to the ambient air.

### Computational method DFT.

The calculation of this work adopted the Vienna ab initio Simulation Package (VASP5.4.4.) based on density functional theory, combined with projector augmented wave (PAW) to describe the core electrons[42,43]. Consequently, the Ba $5s^25p^66s^2$, Ce $5p^66s^24f^15d^1$, Y $4s^24p^65s^24d^1$, Yb $5p^66s^2$ and O $2s^22p^4$ states were treated as valence electrons. The Hubbard $U$ correction was applied to the Ce atoms with the effective $U$ parameter of $U_{eff} = U-J = 5$ eV[44], which reproduces a reasonable band gap[45]. For the optimization of configuration, this work used the electron exchange and correlation within the generalized gradient approximation (GGA) of the Perdew–Burke–Ernzerhof (PBE) functional, the energy cutoff of 500 eV, the $4 \times 4 \times 4$ Gamma-centered k-point mesh, the energy and force convergence criterion for structural relaxation with the value of $1 \times 10^{-5}$ eV and 0.01 eV/Å, respectively. Based on $BaCeO_3$ oxide, the $BaZr_{0.1}Ce_{0.7}Y_{0.1}Yb_{0.1}O_3$ with the lowest total energy is obtained by considering the different occupying positions of Zr, Y and Yb, respectively (Supplementary Fig. 40 and Supplementary Table 6). Then, for the most stable BZCYYb structure, the Y-substitution defect and antisite defect were constructed, respectively, as shown in Supplementary Fig. 41. The diffusion pathways of a proton were calculated with the climbing image nudged elastic band (CI-NEB) method[46,47], from which the energy barrier $E_b$ was achieved.

## Data availability

The data generated in this study are available within the article, Supplementary information, or Source Data file. Source Data are provided with this paper.

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

## Acknowledgements

This work was supported by the National Key R&D Program of China (2022YFB4002502 Jing Zhou. and 2021YFA1502400 L.Z.), the "Transformational Technologies for Clean Energy and Demonstration," Strategic Priority Research Program of the Chinese Academy of Sciences (Grant No. XDA2100000 J.-Q.W.), the National Science Foundation of China (Grant No. 22179141 to L.Z.), DNL Cooperation Fund, CAS (Grant No. DNL202008 to L.Z.) and the Photon Science Center for Carbon Neutrality J.-Q.W.

## Author contributions

L.Z., J.-Q.W., and Z.S. conceived the project and designed the experiments. Z.L. conducted material synthesis, cell fabrication and electrochemical testing. J.C. and Y.Z. contributed to the development of the oxygen electrode. Y.S., Jianqiu Zhu. and C.Z. contributed to the structural characterization and discussion. Jing Zhou. performed the synchrotron experiments. L.L., X.X., and F.C. carried out the DFT calculations. Z.T. and G.K. provided suggestions on the experiments. Z.H. provided an interpretation of XAS and modified the manuscript. Z.L., L.Z., J.-Q.W., and Z.S. wrote the manuscript. All authors discussed the results and commented on the manuscript.

## Competing interests

The authors declare no competing interests.
