## [Peer Review File · Nature Communications]

Reviewers' Comments:

Reviewer #1:

Remarks to the Author:

The authors presented a systematic and comprehensive investigation of the displacement of Y ion during high-temperature sintering. A molten-salt-based synthesis strategy introducing Y into the A site to mitigate ion displacement was demonstrated.

Overall, this is a well-written study in which the authors have given clear and convincing evidence to argue that a solution to prevent structural deterioration during high-temperature sintering due to Y displacement is available. As such, I am convinced that the methodology will impact future electrolyte preparation and protonic fuel cell fabrications.

The attempt to integrate a strong component of experimental investigation with molecular modeling is commendable. The simple DFT calculations do complement experimental characterization well. Still, a deeper theoretical analysis would be desirable. In particular, I remain curious about the mechanism of doping Y at the A site during synthesis that yielded the beneficial effects observed in experiments. Specific questions for the authors to consider:

- Please provide the full meaning of 'SSR'. I cannot find an explanation for this abbreviation.
- I don't see Figure 3 adds much additional information to the article, as it does not contain much detailed information from modeling or experiments. The final structures illustrated for SSR and MSS look almost the same. Is this intended? Back to my earlier comment above, why is the performance so different if the structures are the same?
- Did the authors use DFT+U or just standard DFT in the computational method section? If the latter, the authors should validate their results by including proper U values. This could impact the total energy.
- For Y-substituted lattice calculations, did the authors consider all possible lattice sites? Please clarify this.
- Additional calculations the authors should consider:
 - o Y diffusion from B site to A site
 - o Other possible lattice structures/phases related to A-site substitution.
 - o Proton conduction in MSS structures.
- There are some minor grammar issues and non-standard use of English phrases. Please carefully proofread the manuscript during revision.

This manuscript tells a good story about a particular and relevant problem for protonic fuel cells and is worthy of consideration for publication before serious revisions are completed.

Reviewer #2:

Remarks to the Author:

In this manuscript, the authors present a study on the displacement of Y³⁺ from B site to A site in BaZr_{0.1}Ce_{0.7}Y_{0.1}Yb_{0.1}O_{3-δ} proton-conducting electrolyte during high-temperature firing and the method to solve this problem. The reported results are interesting for the protonic ceramic fuel cell community. However, some revisions are needed to address the comments before considering its publication.

P7: The authors have written that "As shown in Figure 1a, when the samples were quenched from 1300 or 1400 °C, their XRD peaks exhibited obvious shift in position as compared to the pristine sample without the treatment (Figure 1b)".

I assume this is a typing error for Figure 1b, it should be Figure 1a. In addition, the reviewer cannot directly see these changes. Please add the XRD patterns of pristine sample without the treatment for comparison.

P7: "However, when all four samples were annealed at 1300 °C, their diffraction peak positions became similar to each other (Figure 1b)". Please check the Figure 1b, where the composition of the four samples is the same (0.93).

The density of BZCYYb-SSR and BZCYYb-MSS pellet sintered at 1450 °C was characterized by SEM. This method seems somewhat speculative. For example, for the same sample, it is dense in Figure 1h, whereas it is porous in Supplementary Figure 24a. So the density of sintered BZCYYb samples should be measured by the Archimedes method, in order to directly compare the densification of the BZCYYb-SSR and BZCYYb-MSS samples.

The authors have presented the composition of BZCYYb-MSS, $(\text{Ba}_{0.93}\text{Y}_{0.005})(\text{Zr}_{0.1}\text{Ce}_{0.7}\text{Y}_{0.095}\text{Yb}_{0.1})\text{O}_{3-\delta}$, the authors are suggested to give that of the BZCYYb-SSR.

For the BZCYYb-MSS and BZCYYb-SSR pellets sintered at 1450 °C, the authors are recommended to measure their cation compositions, in order to examine the Ba evaporation.

The BZCYYb-MSS and BZCYYb-SSR electrolyte layers are sintered at 1450 °C for applications in PCFCs. The crystal structure stability of BZCYYb-MSS and BZCYYb-SSR quenched at 1300 °C is evaluated using different ways. However, due to the evaporation of Ba, different sintering temperatures will result in different cation compositions, especially the Ba component. Therefore, the Ba compositions of BZCYYb-MSS and BZCYYb-SSR sintered at 1300 and 1450 °C should be different. The authors should measure the crystal structure stability of the samples sintered at 1450 °C, instead of the samples sintered at 1300 °C.

Some minor problems:

ceramic protonic electrolytes, protonic ceramic electrolytes?

In Supplementary Figure 45, Tags of a and b are omitted.

In Figure 4b, adding MSS, in order to compare the SSR in Figure 4c.

Abbreviation, such as YSZ, GDC and PPD, please mark when first used.

Supplementary Figure 20. urier-transformed: typing error.

Reviewer #3:

Remarks to the Author:

The manuscript reports Y ions displacement and the related suppressing technology in BZCYYb electrolyte for PCFC during the preparation process. The densification of protonic electrolyte is a hard issue at present because the contradiction between the high sintering temperature of BZCYYb and the evaporation of Ba and other factors. The experiments were well designed and the results are reliable. However, the discussion and some detail analysis are still required to be improved. Thus it can not be accepted for publication due to the following issues.

1. In Figure 1a, the BZCYYb sample sintered at 1000 °C is not a single phase. In addition, Figure 1b is inconsistent with the description in the manuscript.
2. The manuscript explains that there is Y ion migration during high-temperature sintering of the powders prepared using SSR, which is caused by the volatilization of Ba. Why did the authors choose a Ba-deficient strategy instead of a Ba-rich strategy to address this issue? Would a Ba-rich strategy potentially solve the problem and suppress Y migration?
3. The conductivity of proton conductors is not only related to the intrinsic properties of the materials but also depends on factors such as grain size and grain boundary number. The BZCYYb powders synthesized using the two methods have similar particle sizes, but there is a lack of data comparing grain size and grain boundary properties under the same high-temperature sintering conditions.
4. If the MSS method synthesis can suppress Y migration because Y can occupy the A-site during the synthesis process, can the sol-gel method achieve the same result? The mechanism by which this synthesis method allows Y to occupy the A-site is not clearly explained.
5. The schematic in Figure 3 does not effectively convey the misalignment information of Y ions.
6. Why is there migration of Y ions during high-temperature sintering of BZCYYb instead of Yb ions? The ionic radii of Y and Yb are close, and there are literature reports suggesting that Ce can also occupy the A-site. Could there be migration of Ce ions as well?
7. Why do electrolyte materials synthesized using the SSR method exhibit CO₂ poisoning in a short period of time, whereas this is not observed in materials synthesized using the MSS method? Additionally, the performance of the cells fabricated with SSR-prepared powders significantly

deteriorates in a short time, contrary to other reported results in the literature.

8. From the XPS curves of the electrolyte materials synthesized using the two methods, it appears that the MSS-synthesized material has a lower oxygen vacancy concentration, which seems to be less favorable for proton transport, according to the explanation provided in the paper.

9. The particle morphology of the materials synthesized using the two methods is different, which may have impact on the performance of the materials.

Black text = reviewer' comments

Blue text = our response

Red text = inserted into the revised manuscript

Reviewer #1:

Comments of Reviewer #1:

The authors presented a systematic and comprehensive investigation of the displacement of Y ion during high-temperature sintering. A molten-salt-based synthesis strategy introducing Y into the A site to mitigate ion displacement was demonstrated. Overall, this is a well-written study in which the authors have given clear and convincing evidence to argue that a solution to prevent structural deterioration during high temperature sintering due to Y displacement is available. As such, I am convinced that the methodology will impact future electrolyte preparation and protonic fuel cell fabrications. The attempt to integrate a strong component of experimental investigation with molecular modelling is commendable. The simple DFT calculations do complement experimental characterization well. Still, a deeper theoretical analysis would be desirable. In particular, I remain curious about the mechanism of doping Y at the A site during synthesis that yielded the beneficial effects observed in experiments.

Our response: We would like to thank Reviewer#1 for the positive remarks and useful suggestions to improve quality of our manuscript. We clarify all comments one-by-one below. All the modifications have been highlighted in the revised manuscript.

Comment 1 of Reviewer #1:

Please provide the full meaning of “SSR”. I cannot find an explanation for this abbreviation.

Our response: We would like to thank reviewer#1 for the careful reading and useful suggestion. We have now indicated meaning of SSR in the revised manuscript.

Page 4 in the revised manuscript:

“The as-obtained BZCYYb electrolyte membrane shows high proton conductivity, reaching 4.7×10^{-2} S cm^{-1} at 600 °C, which is 5.2 times that of the sample prepared by the conventional **solid-state reaction (SSR) method.**”

Comment 2 of Reviewer #1:

I don't see Figure 3 adds much additional information to the article, as it does not contain much detailed information from modelling or experiments. The final structures illustrated for SSR and MSS look almost the same. Is this intended? Back to my earlier comment above, why is the performance so different if the structures are the same?

Our response: We appreciate your valuable comments, we apologize that Figure 3 was not clear in the submitted manuscript, we have modified it in the revised manuscript to make difference between SSR and MSS final structures clearer.

For the SSR method (**Figure 3a**), BZCYYb-SSR powder was synthesized at 1000 °C, where the Ba content was 1 as designed, and Y occupied the B site of perovskite. In the high temperature sintering process for preparing PCFCs (1450 °C), Y ion moves from B-to A-site, where Ba evaporated in BZCYYb-SSR with lattice contraction (**Supplementary Table 3**). The severe Ba evaporation and dynamic Y displacement caused a large lattice distortion and strain, thereby resulting in low densification and cracks of electrolyte, harmful to proton transport of electrolyte and PCFCs durability (**Supplementary Figure 43**).

For MSS method (**Figure 3b**), BZCYYb-MSS was prepared at 850°C, and Ba-deficient structure was formed during the synthesis process. In this case, Y was pre-doped into the A-site of perovskite during the powder synthesis process suppressing the Y displacement and structural changes during the high-temperature sintering for preparation of PCFCs. We found enhanced densification (98.4%), smooth surface and large grain size, with an average grain size of 2.31 μm, larger than that of BZCYYb-SSR (1.41 μm), and no cracks in electrolyte, thus enhanced proton transport (**Supplementary Figure 68**). As a result, the BZCYYb-MSS based PCFCs achieved enhanced performance and durability due to the high proton conductivity and crack-free electrolyte.

Figure 3. Schematic diagrams of structural evolution. a Schematic diagram of preparation and structural evolution of BZCYYb-SSR. **b** Schematic diagram of preparation of BZCYYb-MSS and inhibition of Y^{3+} displacement.

Supplementary Table 3. XRD refinement parameters of BZCYYb samples before and after quenching at different temperatures.

Samples	R _p (%)	R _{wp} (%)	R _{exp} (%)	Space group	a (Å)	b (Å)	c (Å)	Volume (Å ³)
BZCYYb-MSS	5.38	9.91	6.04	Imma	6.21	8.74	6.22	338.0
BZCYYb-MSS-1300 °C	9.60	13.63	9.03	Imma	6.21	8.74	6.21	337.5
BZCYYb-MSS-1450 °C	13.18	17.43	7.76	Imma	6.21	8.73	6.21	337.1
BZCYYb-SSR	7.49	10.86	5.56	Imma	6.21	8.81	6.22	340.3
BZCYYb-SSR-1300 °C	4.79	6.37	3.72	Imma	6.15	8.87	6.15	335.8
BZCYYb-SSR-1450 °C	8.03	11.73	4.25	Imma	6.15	8.85	6.15	335.4
BZCYYb-0.98-SSR	11.23	15.63	13.39	Imma	6.21	8.81	6.21	339.6
BZCYYb-0.98-SSR-1300 °C	8.21	11.20	5.42	Imma	6.15	8.86	6.15	335.5
BZCYYb-0.98-SSR-1450 °C	7.46	10.52	5.19	Imma	6.15	8.84	6.15	335.2
BZCYYb-0.95-SSR	11.90	16.94	13.27	Imma	6.20	8.81	6.21	339.5
BZCYYb-0.95-SSR-1300 °C	6.56	9.91	5.53	Imma	6.15	8.87	6.15	335.6
BZCYYb-0.95-SSR-1450 °C	7.94	11.30	5.69	Imma	6.15	8.84	6.15	335.2
BZCYYb-0.93-SSR	12.06	17.25	14.05	Imma	6.17	8.81	6.21	337.4
BZCYYb-0.93-SSR-1300 °C	10.07	12.60	4.50	Imma	6.15	8.71	6.21	335.5
BZCYYb-0.93-SSR-1450 °C	8.40	11.96	4.93	Imma	6.15	8.84	6.16	335.3

Page 44 in the revised manuscript:

Supplementary Figure 43. a, b Surface and c, d cross-sectional SEM images of BZCYYb-SSR membrane sintered at 1450 °C.

Page 69 in the revised manuscript:

Supplementary Figure 68. a, b Surface SEM images of BZCYYb-MSS membrane sintered at 1450 °C.

Comment 3 of Reviewer #1:

Did the authors use DFT+U or just standard DFT in the computational method section? If the latter, the authors should validate their results by including proper U values. This could impact the total energy.

Our response: In the submitted manuscript, standard DFT was applied. Following your suggestion, in the revised manuscript, DFT+U was employed to identify the geometric mechanisms of substitution and proton migration for $\text{BaZr}_{0.1}\text{Ce}_{0.7}\text{Y}_{0.1}\text{Yb}_{0.1}\text{O}_3$. The total energies obtained by DFT and DFT+U calculations for different configurations were compared (**Table 6**). We do see that Hubbard U correction impacts the total energies, but the relative trend remains consistent with the results obtained by standard DFT. Hubbard U correction was included for all calculations in our treatment of the BZCYYb system in this revision (**Supplementary Figure 41, Tables 7-9**).

Supplementary Figure 41. The optimized structural models of (a) BZCYYb, (b) BZCYYb with Y-substitution defect, and (c) BZCYYb with antisite defect.

Supplementary Table 6. The total energies (eV) of different BZCYYb configurations obtained by DFT and DFT+ U methods.

	DFT	DFT+ U
Type 1	-414.67	-396.18
Type 2	-414.67	-396.18
Type 3	-414.67	-396.18
Type 4	-414.67	-396.18
Type 5	-414.66	-396.09
Type 6	-414.62	-396.09
Type 7	-414.65	-396.11
Type 8	/	-396.11
Type 9	/	-396.08
Type 10	/	-396.16

Supplementary Table 7. Configuration parameters of theoretical model of BZCYYb.

BZCYYb-theoretical	Lattice parameter						
	a (Å)	b (Å)	c (Å)	α (°)	β (°)	γ (°)	Volume (Å ³)
	19.42	9.1	6.39	90.3	90	90	1128.48
	M-O bond length (Å)						
	Ba-O7	Ba-O1	Ba-O2	Ba-O3	Ba-O4	Ba-O5	Ba-O6
	2.75	2.70	2.79	3.25	2.88	2.85	2.98

Supplementary Table 8. Configuration parameters of BZCYYb with Y-substitution defect, where Y occupies the Ba site, and the Y original position is empty.

BZCYYb-model 1	Lattice parameter						
	a (Å)	b (Å)	c (Å)	α (°)	β (°)	γ (°)	Volume (Å ³)
	19.45	9.08	6.34	91.2	90.1	89.8	1118.65
	M-O bond length (Å)						
	Y-O7	Y-O1	Y-O2	Y-O3	Y-O4	Y-O5	Y-O6
	2.39	2.45	2.52	2.47	2.33	2.33	2.38

Supplementary Table 9. Configuration parameters of BZCYYb with antisite defect, where Y occupies the Ba site and Ba occupies the Y site.

BZCYYb-model 2	Lattice parameter						
	a (Å)	b (Å)	c (Å)	α (°)	β (°)	γ (°)	Volume (Å ³)
	19.47	9.12	6.38	90.2	90.3	90.1	1132.69
	M-O bond length (Å)						
	Y-O7	Y -O1	Y-O2	Y-O3	Y-O4	Y-O5	Y-O6
	2.24	2.32	2.38	2.81	2.30	2.37	2.50

Page 24 in the revised manuscript:

“The Hubbard U correction was applied to the Ce atoms with effective U parameter of $U_{\text{eff}} = U - J = 5 \text{ eV}$ ⁴³, which reproduce a reasonable band gap⁴⁴.”

Comment 4 of Reviewer #1:

For Y-substituted lattice calculations, did the authors consider all possible lattice sites? Please clarify this.

Our response: Thanks for your comments. In the submitted manuscript, we considered 7 configurations of possible lattice sites, and in the revised manuscript, we have added 3 possible configurations. The 10 possible configurations and corresponding energies are shown in **Supplementary Figure 40** and **Table 6**. And we used the most stable configuration of type 1 for further study.

Supplementary Figure 40. Different configurations of $\text{BaZr}_{0.1}\text{Ce}_{0.7}\text{Y}_{0.1}\text{Yb}_{0.1}\text{O}_3$.

Supplementary Table 6. The total energies (eV) of different BZCYYb configurations obtained by DFT and DFT+*U* methods.

	DFT	DFT+ U
Type 1	-414.67	-396.18
Type 2	-414.67	-396.18
Type 3	-414.67	-396.18
Type 4	-414.67	-396.18
Type 5	-414.66	-396.09
Type 6	-414.62	-396.09
Type 7	-414.65	-396.11
Type 8	/	-396.11
Type 9	/	-396.08
Type 10	/	-396.16

Page 24 in the revised manuscript:

“Based on BaCeO₃ oxide, the BaZr_{0.1}Ce_{0.7}Y_{0.1}Yb_{0.1}O₃ with the lowest total energy is obtained by considering the different occupying positions of Zr, Y and Yb, respectively (**Supplementary Figure 40 and Table 6**).”

Comment 5 of Reviewer #1:

Additional calculations the authors should consider: Y diffusion from B site to A site. Other possible lattice structures/phases related to A-site substitution.

Our response: Thanks for your suggestion. To investigate Y diffusion mechanism, the most stable configuration (Type 1) was chosen as the initial model for further calculations. Following your suggestion, the corresponding structures before and after Y diffusion were further studied to clarify the diffusion of Y from B-site to A-site. Before diffusion, Ba vacancies around Y atom were considered with Y occupied B-site, leading to 6 possible configurations (**Supplementary Figure 42 and Table 10**). After diffusion, Y occupied A-site Ba vacancy, which also leads to 6 possible configurations (**Supplementary Figure 42 and Table 10**). More stable configurations can be found after Y diffusion from B-site to A-site, indicating that it is feasible for Y diffusing from B-to A-site.

Supplementary Figure 42. Possible configurations for Y substituted B-site with Ba vacancy (a-f) and Y substituted A-site (g-l) in BZCYYb.

Supplementary Table 10. The total energies (eV) for different configurations of Y substituted B site and A site in BZCYYb.

	Type 1	Type 2	Type 3	Type 4	Type 5	Type 6
B site	-385.9749	-386.0016	-386.0950	-385.8773	-386.0586	-385.9062
A site	-385.4496	-385.8894	-385.6135	-385.9061	-386.1093	-385.6173

Page 10 in the revised manuscript:

“To clarify Y^{3+} displacement, DFT calculations have also been performed involving the corresponding configurations before and after Y diffusion (**Supplementary Figure 42 and Table 10**). More stable configurations can be found after Y diffusion from B-site to A-site, indicating that the diffusion of Y is favored from thermodynamical view.”

Comment 6 of Reviewer #1:

Proton conduction in MSS structures.

Our response: To evaluate the effect of Y diffusion on the proton migration ability of BZCYYb, we constructed proton adsorption configurations for Y substituted B and A site, respectively (**Supplementary Figures 71, 72**). Besides, the energy barriers (E_b) for proton migration between oxygen sites near the Y dopant were also calculated using the climbing image nudged elastic band method (CI-NEB). Protons follow a 3D diffusion pathway via rotations and jumps between adjacent oxide sites. Rotations are fast, while the jump is considered to be the limiting step. Thus, we only presented the proton jump. The energy barriers (E_b) are 0.34 eV and 0.43 eV for Y substituted B-site (**Supplementary Figure 73**). The energy barrier is 0.33 eV for Y substituted A-site (**Supplementary Figure 74**). Clearly, these energy barriers are all quite small. Therefore, it is expected that proton migration can be achieved for different substitution site of Y.

Supplementary Figure 71. Illustrations of proton positions along different orientations and corresponding proton adsorption energies in BZCYYb for Y substituted B-site.

Supplementary Figure 72. Illustrations of proton positions along different orientations and corresponding proton adsorption energies in B site in BZCYYb for Y substituted A-site.

Supplementary Figure 73. Diffusion path of proton between adjacent lattice oxygen sites and corresponding migration energy barriers for Y substituted B-site in BZCYYb.

Supplementary Figure 74. Diffusion path of proton between adjacent lattice oxygen sites and corresponding migration energy barriers for Y substituted A-site in BZCYYb.

Page 25 in the revised manuscript:

“The diffusion pathways of a proton were calculated with the climbing image nudged elastic band (CI-NEB) method^{45, 46}, from which the energy barrier E_b was achieved.”

Page 16 in the revised manuscript:

“We also performed DFT calculations to investigate the proton transport between Y substituted B-site and A-site (**Supplementary Figures 71, 72**). The energy barriers (E_b) for Y substituted B-site are 0.34 eV and 0.43 eV (**Supplementary Figure 73**), whereas the energy barrier for the Y-substituted A-site is 0.33 eV (**Supplementary Figure 74**). Clearly, these energy barriers are all quite small. Therefore, it is expected that proton migration can be achieved for different substitution site of Y.”

Comment 7 of Reviewer #1:

There are some minor grammar issues and non-standard use of English phrases. Please carefully proofread the manuscript during revision.

Our response: Thanks for your valuable and thoughtful comments. The revised manuscript has been thoroughly checked by the Wiley Editing Services.

Comment 8 of Reviewer #1:

This manuscript tells a good story about a particular and relevant problem for protonic fuel cells and is worthy of consideration for publication before serious revisions are completed.

Our response: Thank you for your positive comments and valuable suggestions to improve the quality of our manuscript.

Reviewer #2:

Comments of Reviewer #2:

In this manuscript, the authors present a study on the displacement of Y^{3+} from B site to A site in $BaZr_{0.1}Ce_{0.7}Y_{0.1}Yb_{0.1}O_{3-\delta}$ proton-conducting electrolyte during high-temperature firing and the method to solve this problem. The reported results are interesting for the protonic ceramic fuel cell community. However, some revisions are needed to address the comments before considering its publication.

Our response: We would like to thank you for the very positive remarks and useful suggestions. We clarify all comments one-by-one below. All the modifications have been highlighted in the revised manuscript.

Comment 1 of Reviewer #2:

P7: The authors have written that “As shown in Figure 1a, when the samples were quenched from 1300 or 1400 °C, their XRD peaks exhibited obvious shift in position as compared to the pristine sample without the treatment (Figure 1b)”.

I assume this is a typing error for Figure 1b, it should be supplementary Figure 1a. In addition, the reviewer cannot directly see these changes. Please add the XRD patterns of pristine sample without the treatment for comparison.

Our response: We would like to thank you for pointing out this typing error, we have corrected it in the revised manuscript. We added the magnified images to show the position shift in **Supplementary Figures 6, 17, 53**. Following your suggestion, we added the XRD patterns of pristine sample without the treatment for comparison in both Figure 1 and Figure 2e.

Figure 1. Study of Structural evolution of BZCYYb prepared by conventional SSR method. a, b XRD patterns of pristine BZCYYb-SSR and BZCYYb-0.93-SSR samples and after quenching at different temperatures.

Supplementary Figure 6. Magnified XRD patterns of BZCYYb-SSR pristine sample and after quenching at different temperatures.

Supplementary Figure 17. Magnified XRD patterns of BZCYYb-0.93-SSR pristine sample and after quenching at different temperatures.

Figure 2. Morphology characterizations of BZCYYb-MSS powder. e XRD patterns of pristine BZCYYb-MSS and after quenching at different temperatures.

Supplementary Figure 53. Magnified XRD patterns of BZCYYb-MSS pristine sample and after quenching at different temperatures.

Page 7 in the revised manuscript:

“...As shown in **Figure 1a** and **Supplementary Figure 6**, when the samples were quenched from **temperatures ≥ 1200 °C**, their XRD peaks exhibit obvious shift in position as compared to the pristine sample without the treatment.”

Comment 2 of Reviewer #2:

P7: “However, when all four samples were annealed at 1300 °C, their diffraction peak positions became similar to each other (Figure 1b)”. Please check the Figure 1b, where the composition of the four samples is the same (0.93).

Our response: We would like to thank you for pointing out this typing error of Figure 1b, we have corrected it to Supplementary Figure 21 in the revised manuscript. The XRD diffraction peaks of four samples including BZCYYb, BZCYYb-0.98, BZCYYb-0.95 and BZCYYb-0.93 prepared by solid state reaction method were shown in **Supplementary Figure 21**.

Supplementary Figure 21. XRD patterns of BZCYYb, BZCYYb-0.98, BZCYYb-0.95 and BZCYYb-0.93 prepared by solid state reaction method after quenching at 1300 °C.

Page 7 in the revised manuscript:

“However, when all four samples were annealed at 1300 °C, their diffraction peak positions become similar to each other (**Supplementary Figure 21**), ...”

Comment 3 of Reviewer #2:

The density of BZCYYb-SSR and BZCYYb-MSS pellet sintered at 1450 °C was characterized by SEM. This method seems somewhat speculative. For example, for the same sample, it is dense in Figure 1h, whereas it is porous in Supplementary Figure 24a. So, the density of sintered BZCYYb samples should be measured by the Archimedes method, in order to directly compare the densification of the BZCYYb-SSR and BZCYYb-MSS samples.

Our response: Following your suggestion, we have measured densification of sintered BZCYYb-SSR and BZCYYb-MSS membranes by Archimedes method. The relative densities of BZCYYb-SSR and BZCYYb-MSS sintered at 1450 °C for 5 h are 97.8 and 98.4%, respectively.

Pages 11 in the revised manuscript:

“From the surface and cross-sectional views of the BZCYYb-SSR pellet, the electrolyte demonstrates rich pores and cracks in the bulk, **showing a relative density of 97.8%.**”

Pages 15 in the revised manuscript:

“As expected, compared to BZCYYb-SSR membranes, the BZCYYb-MSS membranes show enhanced densification (**98.4%**), **smooth surface and large grain size, with an average grain size of 2.31 μm, larger than that of BZCYYb-SSR (1.41 μm), and no cracks (Figures 2k, l and Supplementary Figures 68, 69, 44)**”

Comments 4 and 5 of Reviewer #2:

The authors have presented the composition of BZCYYb-MSS, $(\text{Ba}_{0.93}\text{Y}_{0.005})(\text{Zr}_{0.1}\text{Ce}_{0.7}\text{Y}_{0.095}\text{Yb}_{0.1})\text{O}_{3-\delta}$, the authors are suggested to give that of the BZCYYb-SSR.

For the BZCYYb-MSS and BZCYYb-SSR pellets sintered at 1450 °C, the authors are recommended to measure their cation compositions, in order to examine the Ba evaporation.

Our response: Following your suggestion, we have measured the cation compositions of BZCYYb-SSR and BZCYYb-MSS pellets after sintering at 1450 °C by high-resolution inductively coupled plasma-mass spectrometry (HR-ICP-MS) as shown in **Supplementary Table 11**. And after 1450 °C high temperature sintering, the stoichiometries of Ba in BZCYYb-SSR and BZCYYb-MSS samples were 0.95 and 0.92, respectively.

We also conducted XAS characterizations to examine the coordination configuration of Y cation in BZCYYb-SSR and BZCYYb-MSS samples before and after sintering at 1450 °C (**Figures 1e, 2g**). The EXAFS fitting results demonstrate that the coordination numbers of the Y in the as-synthesized and quenching BZCYYb-SSR are 6 and 6.6, respectively (**Figure 1f, Supplementary Table 5**). And We further calculated the percentage of Y occupying A-and B-sites in perovskite based on the FT-EXAFS fitting data (**Figures 2i and j**). Combined HR-ICP-MS and EXAFS fitting results, the calculated perovskite formula of BZCYYb-SSR is $(\text{Ba}_{0.95}\text{Y}_{0.01})(\text{Zr}_{0.09}\text{Ce}_{0.71}\text{Y}_{0.09}\text{Yb}_{0.1})\text{O}_{3-\delta}$ after sintering at 1450 °C, with 10% of the Y incorporated onto the A-site.

Supplementary Table 11. Cation compositions of BZCYYb-MSS and BZCYYb-SSR powders after quenching at different temperatures from HR-ICP-MS.

	Ba	Ce	Zr	Y	Yb
	Atomic ratio over (Ce + Zr + Y + Yb)				
BZCYYb-MSS-1300 °C	0.92	0.71	0.09	0.11	0.10
BZCYYb-MSS-1450 °C	0.92	0.72	0.08	0.10	0.09
BZCYYb-SSR-1300 °C	0.97	0.72	0.09	0.10	0.10
BZCYYb-SSR-1450 °C	0.95	0.71	0.09	0.10	0.10

Figure 1e Fourier-transformed of Y *K*-edge EXAFS spectra of BZCYYb-SSR before and after quenching at 1450 °C.

Figure 2g Fourier-transformed of Y *K*-edge EXAFS spectra of BZCYYb-MSS before and after quenching at 1450 °C.

Page 14 in the revised manuscript:

“We further measured the cation compositions of BZCYYb-MSS and BZCYYb-SSR after quenching at different temperatures by HR-ICP-MS and calculated the percentage of Y occupying A- and B-sites in perovskite based on the FT-EXAFS fitting data (**Figures 2i, j and Supplementary Table 11**).”

Page 15 in the revised manuscript:

“For the BZCYYb-MSS, the calculated perovskite formula of as-synthesized sample is $(\text{Ba}_{0.93}\text{Y}_{0.005})(\text{Zr}_{0.1}\text{Ce}_{0.71}\text{Y}_{0.095}\text{Yb}_{0.1})\text{O}_{3-\delta}$, where 5% of the Y occupies the A-site. After quenching at 1450 °C, a small increase in the proportion of Y at A-site, about 6.7%, and the formula is $(\text{Ba}_{0.92}\text{Y}_{0.0067})(\text{Zr}_{0.08}\text{Ce}_{0.72}\text{Y}_{0.0933}\text{Yb}_{0.1})\text{O}_{3-\delta}$. As for the BZCYYb-SSR sample, after quenching at 1450 °C, the proportion of Y accommodates on A-site increases significantly from 0 to 10%, and the perovskite formula become $(\text{Ba}_{0.95}\text{Y}_{0.01})(\text{Zr}_{0.09}\text{Ce}_{0.71}\text{Y}_{0.09}\text{Yb}_{0.1})\text{O}_{3-\delta}$.”

Comment 6 of Reviewer #2:

The BZCYYb-MSS and BZCYYb-SSR electrolyte layers are sintered at 1450 °C for applications in PCFCs. The crystal structure stability of BZCYYb-MSS and BZCYYb-SSR quenched at 1300 °C is evaluated using different ways. However, due to the evaporation of Ba, different sintering temperatures will result in different cation compositions, especially the Ba component. Therefore, the Ba compositions of BZCYYb-MSS and BZCYYb-SSR sintered at 1300 and 1450 °C should be different. The authors should measure the crystal structure stability of the samples sintered at 1450 °C, instead of the samples sintered at 1300 °C.

Our response: Following your suggestions, we used HR-ICP-MS to determine the cation compositions of BZCYYb-SSR and BZCYYb-MSS samples sintered at 1300 and 1450 °C, as shown in **Supplementary Table 11**. The stoichiometric contents of Ba in BZCYYb-SSR sintered at 1300 and 1450 °C are 0.97 and 0.95, respectively. And the stoichiometric content of Ba in BZCYYb-MSS sintered at 1300 and 1450 °C is 0.92.

Also, we have performed XRD and TEM characterizations of BZCYYb-SSR (**Figures 1a-c, and Supplementary Figures 8, 9, 12, 13, 16, 20, 26, 28 and Supplementary table 3**) and BZCYYb-MSS (**Figures 2e-f, Supplementary Figures 55, 57 and Supplementary table 3**) samples after quenching at 1450 °C and added them in the revised manuscript and supplementary information. From the XRD refinement results and HR-TEM images, the lattice of BZCYYb-SSR underwent further slight shrinkage after quenching at 1450 °C (relative to 1300°C). For BZCYYb-MSS sample, the XRD diffraction peaks and the lattice spacing of (002) crystal plane hardly changed despite the temperature rising to 1450 °C (**Figure 2f**). These results further suggest that the as-synthesized BZCYYb-MSS sample maintained a stable crystal structure during PCFCs fabrication. We also performed XAS characterizations of BZCYYb-SSR (**Figures 1d-f, Supplementary Figure 36 and Supplementary table 5**) and BZCYYb-MSS (**Figures 2g-h, Supplementary Figures 60 and Supplementary table 5**) samples after quenching at 1450 °C. The EXAFS fitting results demonstrate that the coordination numbers of the Y in BZCYYb-SSR and BZCYYb-MSS samples after the quenching at 1450 °C are 6.4 and 6.6, respectively. And the structural changes are summarized in **Supplementary Tables 3 and 5**.

Supplementary Table 11. Cation compositions of BZCYYb-MSS and BZCYYb-SSR powders after quenching at different temperatures from HR-ICP-MS.

	Ba	Ce	Zr	Y	Yb
	Atomic ratio over (Ce + Zr + Y + Yb)				
BZCYYb-MSS-1300 °C	0.92	0.71	0.09	0.11	0.10
BZCYYb-MSS-1450 °C	0.92	0.72	0.08	0.10	0.09
BZCYYb-SSR-1300 °C	0.97	0.72	0.09	0.10	0.10
BZCYYb-SSR-1450 °C	0.95	0.71	0.09	0.10	0.10

Figure 1. Study of Structural evolution of BZCYYb prepared by conventional SSR method. **a, b** XRD patterns of pristine BZCYYb-SSR and BZCYYb-0.93-SSR samples and after quenching at different temperatures. **c** The change of lattice interplanar spacing of (002) crystal plane before and after quenching at 1450 °C. **d** The spectra of Y K-edge XANES for BZCYYb-SSR before and after quenching. **e** Fourier-transformed of Y K-edge EXAFS spectra of BZCYYb-SSR before and after quenching at 1450 °C. **f** The coordinate number and Y-O bond length for BZCYYb-SSR before and after quenching at 1450 °C.

Supplementary Figure 8. Rietveld refinement plots of the XRD patterns of BZCYYb-SSR quenched at 1450 °C.

Supplementary Figure 9. XRD patterns of pristine BZCYYb-0.98-SSR and quenched at different temperatures.

Supplementary Figure 12. Rietveld refinement plots of the XRD patterns of BZCYYb-0.98-SSR quenched at 1450 °C.

Supplementary Figure 13. XRD patterns of pristine BZCYYb-0.95-SSR and quenched at different temperatures.

Supplementary Figure 16. Rietveld refinement plots of the XRD patterns of BZCYYb-0.95-SSR quenched at 1450 °C.

Supplementary Figure 20. Rietveld refinement plots of the XRD pattern of BZCYYb-0.93-SSR quenched at 1450 °C.

Supplementary Figure 26. TEM images of BZCYYb-SSR powder quenched at 1450 °C. Inset images are the corresponding fast Fourier transform (FFT) pattern of the area in the red box.

Supplementary Figure 28. TEM images of BZCYYb-SSR-0.93 powder quenched at 1450 °C. Inset images are the corresponding fast Fourier transform (FFT) pattern of the area in the red box.

Supplementary Figure 36. Fourier-transformed EXAFS data measured at the Y *K*-edge and its fitting curve for BZCYYb-SSR after quenching at 1450 °C.

Figure 2. Structural characterizations of BZCYYb prepared by MSS method. **e** XRD patterns of pristine BZCYYb-MSS and after quenching at different temperatures. **f** The lattice inter-planar spacing of (002) crystal plane before and after quenching at 1450 °C. **g** Fourier-transformed of Y *K*-edge EXAFS spectra of BZCYYb-MSS before and after quenching at 1450 °C. **h** The coordinate number and Y-O bond length for BZCYYb-SSR before and after quenching at 1450 °C. **i, j** Proportion of Y in sites A and B of BZCYYb-MSS and BZCYYb-SSR samples before and after quenching at 1450 °C.

Supplementary Figure 55. Rietveld refinement plots of the XRD pattern of BZCYYb-MSS quenched at 1450 °C.

Supplementary Figure 57. TEM images of BZCYYb-MSS powder quenched at 1450 °C. Inset images are the corresponding fast Fourier transform (FFT) pattern of the area in the red box.

Supplementary Figure 60. Fourier-transformed EXAFS data measured at the Y *K*-edge and its fitting curve for BZCYYb-MSS after quenching at 1450 °C.

Supplementary Table 3. XRD refinement parameters of BZCYYb samples before and after quenching at different temperatures.

Samples	R _p (%)	R _{wp} (%)	R _{exp} (%)	Space group	a (Å)	b (Å)	c (Å)	Volume (Å ³)
BZCYYb-MSS	5.38	9.91	6.04	Imma	6.21	8.74	6.22	338.0
BZCYYb-MSS-1300 °C	9.60	13.63	9.03	Imma	6.21	8.74	6.21	337.5
BZCYYb-MSS-1450 °C	13.18	17.43	7.76	Imma	6.21	8.73	6.21	337.1
BZCYYb-SSR	7.49	10.86	5.56	Imma	6.21	8.81	6.22	340.3
BZCYYb-SSR-1300 °C	4.79	6.37	3.72	Imma	6.15	8.87	6.15	335.8
BZCYYb-SSR-1450 °C	8.03	11.73	4.25	Imma	6.15	8.85	6.15	335.4
BZCYYb-0.98-SSR	11.23	15.63	13.39	Imma	6.21	8.81	6.21	339.6
BZCYYb-0.98-SSR-1300 °C	8.21	11.20	5.42	Imma	6.15	8.86	6.15	335.5
BZCYYb-0.98-SSR-1450 °C	7.46	10.52	5.19	Imma	6.15	8.84	6.15	335.2
BZCYYb-0.95-SSR	11.90	16.94	13.27	Imma	6.20	8.81	6.21	339.5
BZCYYb-0.95-SSR-1300 °C	6.56	9.91	5.53	Imma	6.15	8.87	6.15	335.6
BZCYYb-0.95-SSR-1450 °C	7.94	11.30	5.69	Imma	6.15	8.84	6.15	335.2
BZCYYb-0.93-SSR	12.06	17.25	14.05	Imma	6.17	8.81	6.21	337.4
BZCYYb-0.93-SSR-1300 °C	10.07	12.60	4.50	Imma	6.15	8.71	6.21	335.5
BZCYYb-0.93-SSR-1450 °C	8.40	11.96	4.93	Imma	6.15	8.84	6.16	335.3

Supplementary Table 5. The structural parameters of the samples derived from R-space fitting curves of EXAFS on Y *K*-edge.

Sample	Bond type	CN*	R (Å)	$\sigma^2 (10^{-3}\text{Å}^2)^{**}$	R factor
Y ₂ O ₃	Y-O	6	2.29±0.02	4.3±1.3	0.001
BZCYYb-MSS	Y-O	6.3±0.7	2.31±0.01	6.8±1.5	0.015
BZCYYb-MSS-1300 °C	Y-O	6.4±0.5	2.29±0.02	7±1.0	0.011
BZCYYb-MSS-1450 °C	Y-O	6.4±0.6	2.29±0.01	6.6±1.3	0.014
BZCYYb-SSR-	Y-O	6.0±0.5	2.33±0.02	6.1±1.3	0.007
BZCYYb-SSR-1300 °C	Y-O	6.6±0.8	2.27±0.02	7.3±0.6	0.004
BZCYYb-SSR-1450 °C	Y-O	6.6±0.7	2.27±0.01	6.6±1.4	0.010

* CN: coordination number; S_0^2 was fixed to be 0.97 obtained from the fitting of Y₂O₃ reference.

** σ^2 : Debye–Waller factors.

Page 7 in the revised manuscript:

“To determine the phase structure of the material at different temperatures, we first annealed the as-synthesized BZCYYb-SSR at certain temperatures (1100, 1200, 1300, **1400 and 1450 °C**) in air for 10 h,”

Page 8 in the revised manuscript:

“The BZCYYb-SSR and BZCYYb-0.93-SSR samples annealed at **1450 °C** show significantly decreased interplanar distances relative to the as-synthesized samples (**Figure 1c, Supplementary Figures 25–28**),”

Page 9 in the revised manuscript:

“According to EXAFS fitting, the average Y-O distance of the BZCYYb-SSR before and after the annealing at 1450 °C is $2.33 \pm 0.01 \text{ \AA}$ vs $2.27 \pm 0.01 \text{ \AA}$, respectively,”

Page 13 in the revised manuscript:

“As seen in **Figure 2e**, the diffraction peaks of BZCYYb-MSS hardly changed despite the temperature rising to 1450 °C.”

Page 14 in the revised manuscript:

“The EXAFS fitting of the BZCYYb-MSS samples before and after quenching at 1450 °C give the Y-O distances of 2.31 ± 0.01 and $2.29 \pm 0.01 \text{ \AA}$, respectively, confirming the negligible lattice shrinkage (**Figures 2g, h and Supplementary Figures 58–60**).”

Page 14 in the revised manuscript:

“The EXAFS fitting results demonstrate that the coordination numbers of the Y in BZCYYb-MSS samples before and after quenching at 1450 °C are 6.3 and 6.4, respectively.”

Comment 7 of Reviewer #2:

Some minor problems:

ceramic protonic electrolytes, protonic ceramic electrolytes?

In Supplementary Figure 45, Tags of a and b are omitted.

In Figure 4b, adding MSS, in order to compare the SSR in Figure 4c.

Abbreviation, such as YSZ, GDC and PPD, please mark when first used.

Supplementary Figure 20. urier-transformed: typing error.

Our response: We carefully checked all context and corrected them in the revised manuscript and supplementary information. We would like to thank you for your careful reading and many useful suggestions.

Page 2 in the revised manuscript:

“However, the practically achievable proton conductivity of **protonic ceramic electrolytes** is still not satisfied due to poor membrane sintering.”

Page 82 in the revised supplementary information.:

Supplementary Figure 81. a Fracture surface of BZCYYb-MSS-based single cell after the 2000 h long-term stability test. **b** High-magnification SEM image of BZCYYb-MSS electrolyte.

Page 18 in the revised manuscript:

Figure 4. Electrochemical performances of PCFC with BZCYYb electrolyte. b *I-V-P* curves of BZCYYb-MSS based single cell.

Page 3 in the revised manuscript:

“... based on oxygen ion-conducting yttrium stabilized zirconia (YSZ) electrolyte^{1, 2, 3, 4.}”

Page 4 in the revised manuscript:

“..., which is 5.2 times that of the sample prepared by the conventional solid-state reaction (SSR) method.”

Page 11 in the revised manuscript:

“..., which is even lower than that of reported oxygen ion conduction electrolyte, such as $\text{Ce}_{0.8}\text{Gd}_{0.2}\text{O}_{1.9}$ (GDC)³⁶ (Figures 1j).”

Page 5 in the revised manuscript:

“The corresponding anode-supported single cell with BZCYYb electrolyte synthesized by the MSS method exhibits low ohmic resistance and distinguished peak power density (PPD) of 964 mW cm^{-2} at 650 °C”

Page 18 in the revised manuscript:

“Moreover, the **open-circuit voltages** (OCVs) of the BZCYYb-MSS-based cell are 1.05, 1.09, 1.10, and 1.11 V at 650, 600, 550, and 500 °C, respectively, which are higher than those of the BZCYYb-SSR-based cell (1.01, 1.05, 1.07, and 1.09 V at 650, 600, 550, and 500 °C, respectively) (**Figures 4b, c).**”

Page 20 in the revised manuscript:

“**Electrochemical impedance spectroscopy (EIS)** was measured for the BZCYYb-MSS-based cell during the long-term stability test (**Figure 4g).**”

Page 36 in the revised supplementary information.:

Supplementary Figure 35. Fourier-transformed EXAFS data measured at the Y K-edge and its fitting curve for BZCYYb-SSR after quenching at 1300 °C.

Reviewer #3:

Comments of Reviewer #3:

The manuscript reports Y ions displacement and the related suppressing technology in BZCYYb electrolyte for PCFC during the preparation process. The densification of protonic electrolyte is a hard issue at present because the contradiction between the high sintering temperature of BZCYYb and the evaporation of Ba and other factors. The experiments were well designed, and the results are reliable. However, the discussion and some detail analyses are still required to be improved. Thus, it can not be accepted for publication due to the following issues.

Our response: We would like to thank you for efforts to review our manuscript and many constructive comments and suggestions, which indeed improve the quality of our manuscript. We clarify all comments one-by-one below. All the modifications have been highlighted in the revised manuscript.

Comment 1 of Reviewer #3:

In Figure 1a, the BZCYYb sample sintered at 1000 °C is not a single phase. In addition, Figure 1b is inconsistent with the description in the manuscript.

Our response: In Figure 1a, we displayed the XRD pattern of the BZCYYb-SSR-pristine sample synthesized at 1000 °C, which is well-matched with the standard pattern BaCeO₃ (PDF# 22-0074) without obvious crystal impurities (**Figure R1**). And we have corrected the incorrect description **Figure 1b** to **Supplementary Figure 6**.

Figure 1. Study of Structural evolution of BZCYYb prepared by conventional SSR method. a XRD patterns of pristine BZCYYb-SSR sample and after quenching at different temperatures.

Figure R1. XRD pattern of pristine BZCYYb-SSR sample synthesized at 1000 °C.

Page 7 in the revised manuscript:

“As shown in **Figure 1a** and **Supplementary Figure 6**, when the samples were quenched from **temperatures ≥ 1200 °C**, their XRD peaks exhibit obvious shift in position as compared to the pristine sample without the treatment.”

Comment 2 of Reviewer #3:

The manuscript explains that there is Y ion migration during high-temperature sintering of the powders prepared using SSR, which is caused by the volatilization of Ba. Why did the authors choose a Ba-deficient strategy instead of a Ba-rich strategy to address this issue? Would a Ba-rich strategy potentially solve the problem and suppress Y migration?

Our response: These are very interesting and professional questions. In the submitted manuscript, we chose the BZCYYb-SSR samples with Ba-deficient in order to determine whether such severe lattice shrinkage of BZCYYb-SSR was caused by the A-site deficiency from Ba evaporation during annealing. Our results show that with or without Ba deficiency, BZCYYYb-SSR exhibited obvious lattice shrinkage after high temperature sintering, suggesting the Ba evaporation may be not the sole reason for the shrinkage of BZCYYb lattice after the sintering at 1300 °C and higher temperatures since all samples possessed different barium contents.

Following your suggestion, we also conducted Ba-rich experiments with the stoichiometric value of Ba is 1.05 to explore the structural stability of BZCYYb (BZCYYb-1.05-SSR). We quenched the as-synthesized BZCYYb-1.05-SSR at 1100, 1200, 1300, 1400 and 1450 °C for 10 h in air to determine the phase structure. As shown in **Supplementary Figure 29**, when the quenching temperature reached to 1200 °C, the XRD peaks of BZCYYb-1.05-SSR exhibited obvious shift in position as compared to the pristine sample, and their diffraction peaks became similar to stoichiometric BZCYYb-SSR (**Figure 1a**). From the XRD refinement results, we confirmed that BZCYYb-1.05-SSR underwent significant lattice shrinkage during high temperature sintering (**Supplementary Figures 30-32** and **Supplementary Table 4**), which was similar to that in BZCYYb-1.0 and BZCYYb-0.93 samples prepared by the SSR method. Therefore, Y migration cannot be inhibited by increasing the stoichiometry of Ba.

Supplementary Figure 29. XRD patterns of pristine BZCYYb-1.05-SSR sample and after quenching at different temperatures.

Supplementary Figure 30. Rietveld refinement plots of the XRD pattern of pristine BZCYYb-1.05-SSR synthesized at 1000 °C.

Supplementary Figure 31. Rietveld refinement plots of the XRD pattern of BZCYYb-1.05-SSR quenched at 1300 °C.

Supplementary Figure 32. Rietveld refinement plots of the XRD pattern of BZCYYb-1.05-SSR quenched at 1450 °C.

Supplementary Table 4. XRD refinement parameters of BZCYYb-1.05-SSR sample before and after quenching.

Samples	R _p (%)	R _{wp} (%)	R _{exp} (%)	Space group	a (Å)	b (Å)	c (Å)	Volume (Å ³)
BZCYYb-1.05-SSR	7.84	10.59	5.68	Imma	6.22	8.80	6.22	340.2
BZCYYb-1.05-SSR-1300 °C	13.40	18.52	14.05	Imma	6.17	8.87	6.15	336.6
BZCYYb-1.05-SSR-1450 °C	6.65	8.96	4.55	Imma	6.16	8.87	6.15	336.2

Page 8 in the revised manuscript:

“We also demonstrated that even after increasing the stoichiometry of Ba to 1.05 (BZCYYb-1.05-SSR), the sample underwent significant lattice shrinkage during high temperature sintering (**Supplementary Figures 29-32 and Supplementary Table 4**), which implies that Y migration cannot be inhibited by increasing the stoichiometry of Ba for BZCYYb-SSR samples.”

Comment 3 of Reviewer #3:

The conductivity of proton conductors is not only related to the intrinsic properties of the materials but also depends on factors such as grain size and grain boundary number. The BZCYYb powders synthesized using the two methods have similar particle sizes, but there is a lack of data comparing grain size and grain boundary properties under the same high-temperature sintering conditions.

Our response: Following your suggestion, we compared the surface grain size of the BZCYYb electrolytes prepared by two different methods after sintering at 1450 °C (**Supplementary Figures 43, 68**). As shown in **Supplementary Figure 68**, the BZCYYb electrolyte synthesized by MSS method shows smooth surface and large grain size, with an average grain size of 2.31 μm (**Supplementary Figures 69**), which is larger than that of BZCYYb-SSR (1.41 μm , **Supplementary Figures 44**). The BZCYYb prepared by the two different methods had similar D50 values, but the BZCYYb-SSR particles were obviously agglomerated with a large particle size distribution range (0-12 μm) (**Supplementary Figures 2, 3**). BZCYYb prepared by molten salt method had regular morphology, well dispersion without agglomeration and large specific surface area, which is conducive to ion diffusion during solid phase sintering, promoting electrolyte densification and reducing the number of grain boundaries.

Supplementary Figure 43. a, b Surface and **c, d** cross-sectional SEM images of BZCYYb-SSR membrane sintered at 1450 °C.

Supplementary Figure 44. Grain size distribution of BZCYYb-SSR electrolyte membrane.

Supplementary Figure 68. a, b Surface SEM images of BZCYYb-MSS membrane sintered at 1450 °C.

Supplementary Figure 69. Grain size distribution of BZCYYb-MSS electrolyte membrane.

Page 15 in the revised manuscript:

“As expected, compared to BZCYYb-SSR membranes, the BZCYYb-MSS membranes show enhanced densification (98.4%), smooth surface and large grain size, with an average grain size of 2.31 μm, larger than that of BZCYYb-SSR (1.41 μm), and no cracks (Figures 2k, l and Supplementary Figures 68, 69, 44)”

Comment 4 of Reviewer #3:

If the MSS method synthesis can suppress Y migration because Y can occupy the A-site during the synthesis process, can the sol-gel method achieve the same result? The mechanism by which this synthesis method allows Y to occupy the A-site is not clearly explained.

Our response: Following your suggestion, we synthesized pure phase BZCYYb by sol-gel method (BZCYYb-SG) and we quenched the as-synthesized BZCYYb-SG at 1100, 1200, 1300, 1400 and 1450 °C for 10 h in air to determine the phase structure (**Supplementary Figure 64**). According to XRD refinements, when the quenching temperature increased to 1300 °C, the cell volume of BZCYYb-SG shrank from 341.0 to 335.4 Å³ (**Supplementary Figures 65, 66 and Supplementary Table 12**). When the quenching temperature further increased to 1450°C, the volume shrank to 333.7 Å³ (**Supplementary Figure 67 and Supplementary Table 12**). From the XRD results, we confirmed that BZCYYb prepared by sol-gel method also underwent significant lattice shrinkage during high temperature sintering. Therefore, we believe that the sol-gel method cannot achieve the same result as the MSS method, that is, Y preferentially occupies the A-site of BZCYYb in the synthesis process.

Compared to conventional solid-state synthesis route, where chemical reactivity is limited by the large diffusion length and slow diffusion of the reacting constituents, the MSS method has lower formation temperature of products as it facilitates fast movement of solids in liquid molten salt phase by means of convection and diffusion. In wet chemical synthesis, solvation is the most crucial step. Most of molecular solvents used in synthesis fail to solvate all metal ions and inorganics. In case of the MSS method, ionic molten salts are used to facilitate chemical reactions at low temperature. Unlike in other synthesis systems, it is possible that most of the metal ions or ionic/covalent bond can get easily destabilized in the MSS process by solvents from the strong polarization induced by molten salt (a pool of ionized cations and anions) at relatively high temperature (*Prog. Mater. Sci.* **2021**, 117, 100734; *Adv. Mater.* **2005**, 17, 2194-2199; *Chem Sci*, **2016**, 7, 855-865; *J. Mater. Sci. Technol.* **2018** 34, 914-930). Three different stages are defined in an MSS process. Stage I involves the mixing of precursors with a salt. The second stage involves the heating of precursor and salt mixture above the melting temperature of the salt to form a molten flux. At this stage, various physical processes take place within the molten salt: uniform dispersion of precursor molecules, dissociation, rearrangement and diffusion. At stage III, nucleation and growth of the product

particles starts via solution-precipitation process. (*Prog. Mater. Sci.* **2021**, 117, 100734; *J Mater Sci Technol.* **2018** 34, 914-930). The special liquid environment enhances the migration and diffusion ability of the reactants, thus achieving atomic-level diffusion. Therefore, we believe that Y pre-doping into the A-site of perovskite can be achieved in molten salts.

Supplementary Figure 64. XRD patterns of pristine BZCYYb-SG sample and after quenching at different temperatures.

Supplementary Figure 65. Rietveld refinement plots of the XRD pattern of the pristine BZCYYb-SG sample.

Supplementary Figure 66. Rietveld refinement plots of the XRD pattern of BZCYYb-SG quenched at 1300 °C.

Supplementary Figure 67. Rietveld refinement plots of the XRD pattern of BZCYYb-SG quenched at 1450 °C.

Supplementary Table 12. XRD refinement parameters of BZCYYb-SG samples before and after quenching.

Samples	R _p (%)	R _{wp} (%)	R _{exp} (%)	Space group	a (Å)	b (Å)	c (Å)	Volume (Å ³)
BZCYYb-SG	7.89	11.45	5.50	Imma	6.21	8.79	6.25	341.0
BZCYYb-SG-1300 °C	7.05	9.74	5.00	Imma	6.15	8.86	6.15	335.4
BZCYYb-SG-1450 °C	7.69	10.60	4.38	Imma	6.15	8.84	6.15	335.2

Page 15 in the revised manuscript:

“And we also demonstrated that BZCYYb prepared by common sol-gel method, denoted as BZCYYb-SG, could not inhibit Y migration (**Supplementary Figures 64–67**, **Supplementary Table 12**).”

Comment 5 of Reviewer #3:

The schematic in Figure 3 does not effectively convey the misalignment information of Y ions.

Our response: Considering your remark, we modified the diagram of Y ions misalignment information to make it clear.

Page 17 in the revised manuscript:

Figure 3. Schematic diagrams of structural evolution. a Schematic diagram of preparation and structural evolution of BZCYYb-SSR. **b** Schematic diagram of preparation of BZCYYb-MSS and inhibition of Y^{3+} displacement.

Comment 6 of Reviewer #3:

Why is there migration of Y ions during high-temperature sintering of BZCYYb instead of Yb ions? The ionic radii of Y and Yb are close, and there are literature reports suggesting that Ce can also occupy the A-site. Could there be migration of Ce ions as well?

Our response: These are all relevant and interesting questions. According to your comments, BaZr_{0.1}Ce_{0.7}Yb_{0.2}O_{3-δ} (BZCYb0.2) sample with Yb content of 20% was prepared by solid state reaction method to exclude the effect of Y on the structure changes. For the BZCYb0.2 sample, we also quenched the as-synthesized BZCYb0.2 at 1100, 1200, 1300, 1400 and 1450 °C for 10 h in air to determine the phase structure (**Supplementary Figure 37**). According to XRD results, when the quenching temperature lower than 1300 °C, the position of the XRD diffraction peak did not change obviously. And when the quenching temperature increased to 1400 °C, the diffraction peak shift to the right, as shown in **Supplementary Figure 38**, and a secondary phase of Yb₂O₃ was detected. The ionic radius of Yb³⁺ is slightly smaller than that of Y³⁺ (0.868 Å vs 0.9Å) and closer to that of Ce⁴⁺ (0.87 Å). It can be inferred from the XRD results that Yb preferentially precipitates in the form of secondary phase Yb₂O₃ during high temperature sintering process.

To investigate the possibility of Ce migration, X-ray absorption near-edge structure (XANES) analysis was performed at the Ce L₃ edge of all BZCYYb samples as shown in **Supplementary Figure 61**, along with reference CeO₂ with local 8-fold coordination and SrCeO₃ with local 6-fold coordination. All feature and energy position of the Ce-L₃ XANES spectrum of BZCYYb before and after quenching at 1300 °C are almost same to those of SrCeO₃, confirming that Ce exists in a Ce⁴⁺ state with local octahedral coordination in all BZCYYb samples and exists in the B site of perovskite. The splitting of each peak at the Ce-L₃ edge due to 5d crystal field splitting reduces with increase in coordination number from CeO₆ in SrCeO₃ to CeO₈ in CeO₂¹, and will be further narrower at the A site for CeO₁₂. Therefore, for both BZCYYb-MSS and BZCYYb-SSR samples, we can exclude meaningful amount of Ce ions at the A site.

Supplementary Figure 37. XRD patterns of pristine BZCYb0.2-SSR sample and after quenching at different temperatures.

Supplementary Discussion: $\text{BaZr}_{0.1}\text{Ce}_{0.7}\text{Yb}_{0.2}\text{O}_{3-\delta}$ sample with Yb content of 20% was prepared by solid state reaction method (BZCYb0.2-SSR), and the treatments at the same temperatures as BZCYYb-SSR were conducted (**Supplementary Figure 37**). However, when the temperature reached 1400 °C, some impurities were detected, and the impurity phase was Yb_2O_3 (**Supplementary Figure 38**). This result indicated that Yb^{3+} preferentially precipitates in the form of secondary phase Yb_2O_3 during high temperature sintering process.

Supplementary Figure 38. XRD pattern of pristine BZCYb0.2-SSR sample after quenching at 1400 °C.

Supplementary Figure 61. The spectra of Ce L_3 edge XANES for BZCYYb-MSS and BZCYYb-SSR before and after quenching at 1300 °C.

Page 9 in the revised manuscript:

“It was observed that Yb^{3+} was preferentially precipitated in the form of Yb_2O_3 phase during high temperature sintering process (**Supplementary Figures 37, 38**).”

Page 14 in the revised manuscript:

“And we proved that the fine structure of Ce did not change before and after quenching (**Supplementary Figure 61**).”

Page 62 in the revised supplementary information:

Supplementary Discussion: To investigate the possibility of Ce migration, X-ray absorption near-edge structure (XANES) analysis was performed at the Ce L_3 edge of all BZCYYb samples as shown in **Supplementary Figure 61**, along with reference CeO_2 with local 8-fold coordination and SrCeO_3 with local 6-fold coordination. All feature and energy position of the Ce- L_3 XANES spectrum of BZCYYb before and after quenching at 1300 °C are almost same to those of SrCeO_3 , confirming that Ce exists in a Ce^{4+} state with local octahedral coordination in all BZCYYb samples and exists in the B site of perovskite. The splitting of each peak at the Ce- L_3 edge due to 5d crystal field splitting reduces with increase in coordination number from CeO_6 in SrCeO_3 to CeO_8 in CeO_2 ¹, and will be further narrower at A site for CeO_{12} . Therefore, for both BZCYYb-MSS and BZCYYb-SSR samples, we can exclude meaningful amount of Ce ions at the A-site.

Comment 7 of Reviewer #3:

Why do electrolyte materials synthesized using the SSR method exhibit CO₂ poisoning in a short period of time, whereas this is not observed in materials synthesized using the MSS method? Additionally, the performance of the cells fabricated with SSR-prepared powders significantly deteriorates in a short time, contrary to other reported results in the literature.

Our response: Thanks for your comments. The Ba content of BZCYYb-MSS was smaller than the designed stoichiometric (0.93 vs 1) as confirmed by the EDS mapping and HR-ICP-MS results, which reduced the chemical potential of the basic A-site cation in perovskite oxides and improved the stability towards carbonation or hydration. Other work confirmed that the structural stability and CO₂ resistance of the material can be significantly improved by introducing Ba defects (*J. Power Sources* **2021**, 493 229691, *Int. J. Hydrog. Energ.* **2015**, 40, 11022-11031, *Solid State Ion.* 1998, **110** 103–110).

XPS spectra indicated that the oxygen vacancy concentration of BZCYYb-MSS is lower than that of BZCYYb-SSR (**Supplementary Figures 39 and 62**). And the oxygen vacancy on the metal oxide is generally accepted as the absorption site of CO₂, and the oxygen vacancy can interact with nonbonding electrons from the O atoms in CO₂ via the Lewis acid–base interaction due to its Lewis acid nature. (*ACS Catal.* **2020**, 10, 19, 11493–11509). Moreover, several DFT calculations have indicated the assistance of surface oxygen vacancy in CO₂ adsorption and activation (*ACS Catal.* **2013**, 3, 1296-1306; *Appl. Catal. A* **2016**, 521, 240–249; *RSC Adv.*, **2015**, 5, 97528-97535). And BZCYYb-MSS samples exhibited higher crystallinity and lower oxygen vacancy concentration, and the reduced oxygen vacancy concentration indicates the enhancement of ordering in lattice, thus making the structure more stable (*Int. J. Hydrog. Energ.* **2015**, 40, 11022-11031). For BZCYYb-SSR sample, high-temperature sintering lead to the evaporation of Ba, which may partly be located in the grain boundary region of the sample and eventually react with atmospheric carbon dioxide.

The cathode was exposed to static air without gas purges, and water vapor was generated on the cathode side during the PCFCs operation, so the electrolyte on the cathode side was exposed to an atmosphere

containing CO₂ and water vapor. For the BZCYYb-SSR electrolyte, the dynamic displacement of Y³⁺ during the high-temperature sintering process led to lattice distortion and strain, resulting in more pore structures and grain boundaries in the electrolyte. It is reported that the grain boundaries and precipitated Ba are the penetration path of water vapor and CO₂ and corrode the interior of the electrolyte along the grain boundaries, destroying the electrolyte structure, facilitating chemical decomposition, leading to mechanical disintegration, and ultimately resulting in the attenuation of cell performance (*Nano Lett.* **2018**, 18, 1110-1117). Although the reaction at the topmost surface is unavoidable, the highly densified microstructure and stable lattice structure provides the significant advantage of chemical stability as well as proton conduction.

Comment 8 of Reviewer #3:

From the XPS curves of the electrolyte materials synthesized using the two methods, it appears that the MSS-synthesized material has a lower oxygen vacancy concentration, which seems to be less favourable for proton transport, according to the explanation provided in the paper.

Our response: According to the hydration mechanism, the moderate oxygen vacancy concentration is one of the conditions that affect proton uptake, so in the atmosphere containing water vapor, the oxygen vacancy concentration directly affects the proton uptake and proton concentration. Grotthuss and Vehicle mechanisms are well-known proton conducting mechanisms in oxides (**Figure R2**). The activation energy of Vehicle (E_a) (>0.4 eV) is higher than Grotthuss (E_a) (<0.4 eV) owing to the low mobility of O^{2-} due to its larger size. Therefore, due to the lower E_a , the latter Grotthuss mechanism has been widely accepted in proton-conducting oxides (*Adv. Energy Mater.* **2022**, 12, 2201882; *J. Phys. Energy.* **2021**, 3 032019). And the existence of oxygen vacancies in the perovskites would have increased the distance of protons jumping from one oxygen atom to the other and thus decreased proton conductivity (*J. Phys. Chem. C* **2020**, 124, 8024–8033). The activation energies of several proton transfer pathways for protons in BZCYYb are shown in **Figure R3** (*J. Phys. Chem. C* **2020**, 124, 8024–8033). Therefore, oxygen vacancy can improve the proton absorption capacity, but the effects on the proton transport are complicated.

In addition, oxygen vacancy is not the determining factor for proton transport, which is also related to the density of electrolyte film, grain size and grain boundary number. The BZCYYb-MSS electrolyte showed high relative density with large grain size. However, for BZCYYb-SSR electrolyte, Y cation dynamic migration from B-site to A-site consumes oxygen vacancy concentration in the BZCYYb-SSR sample during high-temperature sintering, which lead to large lattice distortion and strain, thus leading to low densification and cracking of electrolyte, harmful to proton transport.

Figure R2. Models of proton conduction. a) Grotthuss, b) Vehicle mechanisms.

Figure R3. Selected representative paths of proton transfer in BZCYYb.

Comment 9 of Reviewer #3:

The particle morphology of the materials synthesized using the two methods is different, which may have impact on the performance of the materials.

Our response: Considering your statement, we have determined the specific surface area of BZCYYb-SSR and BZCYYb-MSS samples by the Brunauer-Emmett-Teller (BET) method. As shown in **Supplementary Figures 4 and 50**, the specific surface area of BZCYYb-MSS is $3.3617 \text{ m}^2 \text{ g}^{-1}$, larger than that of BZCYYb-SSR ($1.8967 \text{ m}^2 \text{ g}^{-1}$). It is well known that in the solid sintering mechanisms, the substance transport rate strongly depends on the particle size and surface area, and the sintering temperature decreases greatly with the decrease of particle size. Compared with BZCYYb-SSR, the average particle size of BZCYYb-MSS is smaller, and well dispersed BZCYYb-MSS particles have narrow particle size distribution and larger specific surface area, which benefits to solid-state diffusion, further improving densification of electrolyte pellets, reducing the grain boundary, and promoting proton transport.

Supplementary Figure 4. The N₂ adsorption-desorption isotherm of pristine BZCYYb-SSR powder.

Supplementary Figure 50. The N₂ adsorption-desorption isotherm of pristine BZCYYb-MSS powder.

Page 5 in the revised manuscript:

“Moreover, the scanning electron microscopy (SEM) images and particle-size distribution of BZCYYb-SSR reveals irregular and significantly agglomerated particles with an average particle size of 1.95 μm and specific surface area of 1.90 m² g⁻¹ (**Supplementary Figures 2 - 4**).”

Page 11 in the revised manuscript:

“**Figure 2a** indicates that the BZCYYb-MSS synthesized using a salt-to-nitrate ratio of 2/1 shows the most uniform particle-size distribution among all the samples, with an average particle size of 1.86 μm and specific surface area of 3.36 m² g⁻¹ (**Supplementary Figures 48–50**)”

Reviewers' Comments:

Reviewer #2:

Remarks to the Author:

The authors have addressed my questions. I recommend accepting this manuscript for publication.

Reviewer #3:

Remarks to the Author:

It has addressed all the questions from the reviewers. No further revision is required.

■ Point-by-point response to the reviewers' comments

Reviewer #2:

The authors have addressed my questions. I recommend accepting this manuscript for publication.

Our response: We would like to thank you for the very positive remarks.

Reviewer #3:

It has addressed all the questions from the reviewers. No further revision is required.

Our response: We would like to thank you for the very positive remarks.